# HUMAN-LLM COLLABORATIVE FEATURE ENGINEERING FOR TABULAR LEARNING

**Zhuoyan Li[1], Aditya Bansal[2], Jinzhao Li[1], Shishuang He[3], Zhuoran Lu[1],**
**Mutian Zhang[1], Yiwei Yang[4], Qin Liu[5], Swati Jain[2], Ming Yin[1], Yunyao Li[2]**
[1]Purdue University    [2]Adobe    [3]UIUC    [5]University of Washington    [4]UC Davis

## ABSTRACT

Large language models (LLMs) are increasingly used to automate feature engineering in tabular learning. Given task-specific information, LLMs can propose diverse feature transformation operations to enhance downstream model performance. However, current approaches typically assign the LLM as a black-box optimizer, responsible for both proposing and selecting operations based solely on its internal heuristics, which often lack calibrated estimations of operation utility and consequently lead to repeated exploration of low-yield operations without a principled strategy for prioritizing promising directions. In this paper, we propose a human–LLM collaborative feature engineering framework for tabular learning. We begin by decoupling the transformation operation proposal and selection processes, where LLMs are used solely to generate operation candidates, while the selection is guided by explicitly modeling the utility and uncertainty of each proposed operation. Since accurate utility estimation can be difficult especially in the early rounds of feature engineering, we design a mechanism within the framework that selectively elicits and incorporates human expert preference feedback—comparing which operations are more promising—into the selection process to help identify more effective operations. Our evaluations on both the synthetic study and the real user study demonstrate that the proposed framework improves feature engineering performance across a variety of tabular datasets and reduces users' cognitive load during the feature engineering process.

## 1 INTRODUCTION

Today, AI-powered models are increasingly integrated into diverse applications, such as fraud detection (Yang et al., 2025) and online platform recommendation (Alfaifi, 2024; Resnick & Varian, 1997), where tabular data remains one of the most popular data modalities to represent structured information like financial transaction logs or user profiles Nam et al. (2023). The quality of the feature engineering, which transforms raw feature columns into meaningful representations, often plays a critical role in determining the performance of AI models. To reduce manual effort in the feature engineering, traditional AutoML-based methods (Zhang et al., 2023; Erickson et al., 2020) have been developed to automate feature construction by applying predefined transformation operators over feature columns, which often lack the task-specific understanding and tend to generate redundant features that lead to suboptimal model performance.

Recent advances in large language models (LLMs) (Achiam et al., 2023), with their strong language understanding and reasoning abilities, have motivated numerous studies to directly apply LLMs in feature engineering for tabular prediction tasks, aiming to leverage their semantic understanding to create informative features to improve the downstream model performance (Bordt et al., 2024; Han et al., 2025). For example, CAAFE (Hollmann et al., 2023) uses the task description together with the semantic description of each feature to prompt an LLM to iteratively synthesize semantically meaningful features. Nam et al. treats the LLM as a black-box optimizer that proposes and refines feature generation rules using feedback from the validation score of the downstream AI model and the verbalized reasoning distilled from a fitted decision tree. Although these approaches can improve performance, the current approaches typically assign the LLM both the role of proposing feature transformation operations and the role of selecting among them, so that the entire process is driven purely by the model's internal heuristics, which often lacks calibrated estimates of the utility and the

uncertainty of each operation and in turn can lead to repeated exploration of low-yield operations without a principled strategy for prioritizing more promising directions. As a result, it tends to waste evaluation budget on low-yield transformations and performs poorly when the number of feature engineering iterations is limited.

In this paper, we propose a human–LLM collaborative feature engineering framework for tabular learning. We begin by decoupling the transformation operation proposal and selection processes, where LLMs are used solely to propose diverse transformation operation candidates based on their internal heuristics and understanding of the current task, while the selection is guided by explicit modeling of the utility and uncertainty of each proposed operation. However, the estimated utility of feature operations can often be poorly calibrated against their actual utility, particularly in the early rounds of feature engineering when only limited observational data is available to fit the estimation model. In such cases, human expert collaborators (e.g., machine learning practitioners), who have accumulated domain knowledge about which types of feature transformations are likely to be beneficial, can provide qualitative insights to support more informed selection among the LLM-proposed feature operations. To incorporate such qualitative knowledge into the selection process in a tractable manner, following prior work (AV et al., 2022; Xu et al., 2024a), we consider the form of *preference feedback* [1] from the human, where they compare pairs of feature operations to indicate which is better. To effectively leverage this form of human preference feedback without incurring excessive human cognitive cost, we design a mechanism within the framework that selectively elicits preference feedback from the human collaborator only when the potential gain from the human feedback can justify this additional human effort. To evaluate the effectiveness of the proposed framework, we first conducted a synthetic study across a variety of tabular datasets. Compared to different baselines including both AutoML methods and LLM-powered feature engineering methods, our method exhibits consistent improvement when evaluated with different downstream models. We then conducted a user study to further understand how actual users collaborate with the LLM in our framework. We observed that our algorithm can also improve the actual feature engineering performance and reduce the cognitive load experienced by human experts during the process.

## 2 RELATED WORK

***Large Language Models (LLMs) for Tabular Learning.*** Recent advances in large language models (LLMs) (Achiam et al., 2023), with their strong language understanding and reasoning abilities, have motivated numerous studies to apply LLMs to tabular prediction tasks (Wang et al., 2023; Ko et al., 2025; Bouadi et al., 2025; Zhang et al., 2024; Han et al., 2024; Nam et al., 2024b). One line of work adapts LLMs by converting structured and tabular data into natural language and then leveraging zero- or few-shot inference or fine-tuning of LLMs (Dinh et al., 2022; Hegselmann et al., 2023; Yan et al., 2024). More recently, researchers have explored using LLMs to directly propose and select new features from the original feature columns and task descriptions to augment predictive performance (Hollmann et al., 2023; Bordt et al., 2024; Han et al., 2025; Abhyankar et al., 2025). For example, Nam et al. treats the LLM as a black-box optimizer which proposes and refines feature generation rules using feedback from the validation score of the downstream machine learning model and the verbalized reasoning distilled from a fitted decision tree. In this paper, we ask whether it is possible to decouple the generation and the selection process by explicitly modeling both the utility and the uncertainty of LLM-generated feature operations with Bayesian optimization, so as to guide the selection and composition of features in a more efficient and principled manner.

***Human-AI Collaboration for Interactive Machine Learning.*** There has been a growing interest among researchers in leveraging human knowledge to enhance standard machine learning pipelines. Many studies investigate the factors that influence effective human–AI collaboration (Lai et al., 2021; Buçinca et al., 2021; Chiang et al., 2023; 2024). For example,Wang et al. conducted interviews with industry practitioners to understand their experiences with AutoML systems. Some recent work has focused on redesigning the interaction between humans and models during the learning process (Mozannar et al., 2023; Wei et al., 2024; De Toni et al., 2024; Alur et al., 2024). In this line of research, collaboration is operationalized through mechanisms such as modifying the objective functions to promote human-AI complementarity (Bansal et al., 2019a;b; Mahmood et al.,

---

[1] As shown in prior research (Kahneman & Tversky, 2013), this pairwise preference format enables humans to more effectively express their internal judgments compared to directly evaluating individual instances.

2024), adjusting how to present information like AI confidence or explanation based on the human behavior modeling (Vodrahalli et al., 2022; Li & Yin, 2024; Li et al., 2024; Lu et al., 2023; Li et al., 2023b; 2025), and reducing human workload by shifting their role from primary decision maker to on-demand advisor, typically through selective querying paradigms that solicit human input only when it is expected to be most valuable (AV et al., 2022; Xu et al., 2024a;b; Souza et al., 2021; Hvarfner et al., 2022). In this paper, we ask how human expertise should inform and shape the LLM-powered feature engineering process in order to achieve more complementary outcomes.

## 3 METHODOLOGY

### 3.1 MOTIVATION AND PROBLEM FORMULATION

In this study, we explore the scenario of human–LLM collaborative feature engineering in supervised tabular prediction tasks, and we now formally describe it. Let $\mathcal{D} = \{(\boldsymbol{x}_i, y_i)\}_{i=1}^n$ denote the tabular dataset, where each $\boldsymbol{x}_i \in \mathbb{R}^d$ is a $d$-dimensional feature vector for the task instance, and the $j$-th dimension corresponds to a feature column with name $c_j \in C = \{c_1, ..., c_d\}$. The label $y_i$ is the target output, with $y_i \in \{0, 1, ..., K\}$ for a $K$-class classification task and $y_i \in \mathbb{R}$ for a regression task. By splitting the dataset $\mathcal{D}$ into the training set $\mathcal{D}_{\text{train}}$ and the validation set $\mathcal{D}_{\text{val}}$, a tabular learner $f_{\text{tabular}}$ is trained on $\mathcal{D}_{\text{train}}$ and evaluated on $\mathcal{D}_{\text{val}}$ by a task-appropriate score function $J(f_{\text{tabular}}; \mathcal{D}_{\text{val}})$ (e.g., AUCROC for classification task, and negative MSE for regression task). Let $\Phi$ denote the space of candidate feature transformation operations. Each operation $e \in \mathcal{E}$ maps the original data matrix $\mathcal{X} \in \mathbb{R}^{n \times d}$ into a new feature column $z_e \in \mathbb{R}^n$. The updated training and validation datasets are represented as $\mathcal{D}_{\text{train}} \oplus e$ and $\mathcal{D}_{\text{val}} \oplus e$, respectively, where $\oplus$ represents the addition of the new feature column $z_e$ to the existing dataset. To evaluate the utility of a feature operation $e$, we first define the black-box utility function $g(\cdot)$:

$$g(e) = J(f_{\text{tabular}}; \mathcal{D}_{\text{val}} \oplus e), \quad \text{where} \quad f_{\text{tabular}} = \arg\min_f \mathcal{L}(f; \mathcal{D}_{\text{train}} \oplus e) \tag{1}$$

Here, $\mathcal{L}$ is the loss function used to train the tabular learner (e.g., cross-entropy for classification or MSE for regression). Since feature engineering proceeds in multiple rounds, where at each round $t$, we evaluate the utility of a selected transformation $e_t$, and update $\mathcal{D}_{\text{train}}$ and $\mathcal{D}_{\text{val}}$ only if $g(e_t) > 0$. The goal of feature engineering in each round is to identify the operation that maximizes predictive performance on the current validation set :

$$e_t^\star = \arg\max_{e \in \mathcal{E}} g(e) \tag{2}$$

However, the utility function $g(\cdot)$ is observable only after an expensive refit and re-evaluate cycle on the updated model. Prior work on LLM-powered feature engineering (Hollmann et al., 2023; Nam et al., 2024a) addresses this black-box optimization by treating the LLM as an implicit surrogate for $g(\cdot)$, with the LLM $M$ handling both proposal and selection of the optimal transformation $e^\star$. In round $t$ of feature engineering process, given the observed performance history $H_t = \{(e_i, g(e_i))\}_{i=1}^{t-1}$, column descriptions $C$, and dataset-level metadata including the prediction objective Meta, the LLM samples a set of $N$ candidates $\mathcal{S}_t = \{e_t^1, ..., e_t^N\}$ from its internal proposal distribution:

$$\mathcal{S}_t \sim \mathcal{P}_M(\cdot \mid H_t, C, \text{Meta}) \tag{3}$$

The LLM $M$ would then select a candidate $e_t \in \mathcal{S}_t$ based on internal heuristics about which transformation might be most useful in the current round. In this paper, we ask whether it is possible to decouple transformation operation proposal and selection processes, where the LLM $M$ serves solely as a operation proposal generator, sampling candidates from the distribution $\mathcal{P}_M(\cdot \mid H_t, C, \text{Meta})$, and the selection is guided by explicitly modeling the utility of LLM-proposed feature operations. To address the black-box nature of the utility function $g(\cdot)$, we adopt a Bayesian optimization approach that first constructs an explicit surrogate model $\hat{g}(\cdot)$ based on the history $H_t$ to estimate the utility and uncertainty of each operation. Based on this surrogate model, we next move to explore how to select among the LLM-proposed candidate operations in each round, and how to selectively elicit and incorporate human preference feedback into the model selection process.

### 3.2 SURROGATE MODEL FOR APPROXIMATION OF THE UTILITY FUNCTION $g(\cdot)$

In this part, we describe how to construct a surrogate model $\hat{g}(\cdot)$ to approximate $g(\cdot)$.

*Encoding of LLM-Proposed Feature Operations.* To allow the surrogate model to effectively approximate the utility function $g(\cdot)$, each LLM-proposed feature transformation $e$ is first mapped into a vector representation. We first apply a pretrained embedding encoder $\phi_{\text{embedding}}(e)$ [2] to obtain a dense semantic representation of the operation $e$, which captures compositional and linguistic relationships among candidate operations that are learnable by the surrogate model. However, the semantic embedding $\phi_{\text{embedding}}(e)$ alone may be insufficient when multiple feature columns have similar linguistic descriptions. To address this, we incorporate a column-usage encoder $\phi_{\text{column}}(e)$ to provide explicit structural information about which feature columns are used in the feature operation. Specifically, $\phi_{\text{column}}(\cdot)$ maps an operation $e$ into a binary vector $\boldsymbol{m} \in \{0,1\}^d$, where $\boldsymbol{m} = \big[\, \mathbb{I}[\, c_i \text{ is used in } e\,]\,\big]_{i=1}^d$ and each $c_i \in C$ denotes a column from the input data matrix. Finally, the overall embedding representation for the surrogate model $\hat{g}$ is obtained by concatenating the two components:

$$\phi(e) \;=\; \big[\, \phi_{\text{embedding}}(e), \phi_{\text{column}}(e)\,\big] \tag{4}$$

*Bayesian Neural Network as Surrogate Model.* In Bayesian optimization, Gaussian processes (GPs) are widely used as surrogate models in different tasks such as hyperparameter tuning (Snoek et al., 2012) and system design (Wang et al., 2024), which typically involve relatively simple, low-dimensional feature spaces that GPs can scale effectively (Xu et al., 2024c). In contrast, our setting requires modeling LLM-proposed feature operations that are expressed in natural-language format and mapped via an encoder $\phi(\cdot)$ into a high-dimensional representation, where GPs often struggle to scale and capture non-stationarity (Snoek et al., 2015). Therefore, in this paper, we opted for Bayesian neural network (BNN) as the surrogate $\hat{g}$, which can provide greater scalability and expressiveness for modeling high-dimensional, language-derived feature embeddings (Li et al., 2023a). Specifically, given the performance history at the current round $H_t = \{(e_i, g(e_i))\}_{i=1}^{t-1}$, the surrogate $\hat{g}$ is constructed as a Bayesian neural network parameterized by $\boldsymbol{\theta}$, which defines a predictive model $\hat{g}(\phi(e); \boldsymbol{\theta})$ over candidate operations $e$ to approximate their true utility $g(e)$. To capture uncertainty, instead of learning a single point estimate of $\boldsymbol{\theta}$, we adopt a Bayesian approach and set out to learn the posterior distribution of the model parameters conditioned on history $H_t$, i.e., $\mathcal{P}(\boldsymbol{\theta} \mid H_t)$. As directly computing this posterior $\mathcal{P}(\boldsymbol{\theta} \mid H_t)$ is intractable, we learn the variational distribution $q_t(\boldsymbol{\theta}) = \mathcal{N}(\boldsymbol{\theta}; \boldsymbol{M}_t, \boldsymbol{\Sigma}_t)$ by minimizing the KL divergence between $q_t(\boldsymbol{\theta})$ and the true posterior:

$$\begin{aligned}
\mathrm{KL}(q_t(\boldsymbol{\theta})\|\mathcal{P}(\boldsymbol{\theta} \mid H_t)) &= \int q_t(\boldsymbol{\theta}) \log \frac{q_t(\boldsymbol{\theta})}{\mathcal{P}(\boldsymbol{\theta} \mid H_t)}\, d\boldsymbol{\theta} \\
&= \int q_t(\boldsymbol{\theta}) \Big( \log \tfrac{q_t(\boldsymbol{\theta})}{\mathcal{P}(\boldsymbol{\theta})} - \log \mathcal{P}(H_t \mid \boldsymbol{\theta}) + \log \mathcal{P}(H_t)\Big) d\boldsymbol{\theta} \\
&= \mathrm{KL}(q_t(\boldsymbol{\theta})\|\mathcal{P}(\boldsymbol{\theta})) - \mathbb{E}_{q_t(\boldsymbol{\theta})}\big[ \log \mathcal{P}(H_t \mid \boldsymbol{\theta}) - \log \mathcal{P}(H_t)\big]
\end{aligned} \tag{5}$$

where $\mathcal{P}(\boldsymbol{\theta})$ is the prior distribution over model parameters and $\mathcal{P}(H_t)$ is a constant[3]. Given the learned variational posterior $q_t(\boldsymbol{\theta})$, the predicted *expected utility* $\mu_t(e)$ of a candidate operation $e$ and its corresponding *uncertainty* $\sigma_t^2(e)$ are computed as:

$$\mu_t(e) = \mathbb{E}_{q_t(\boldsymbol{\theta})}[\hat{g}(\phi(e); \boldsymbol{\theta})], \quad \sigma_t^2(e) = \mathbb{E}_{q_t(\boldsymbol{\theta})}\big[\hat{g}(\phi(e); \boldsymbol{\theta})^2\big] - \mu_t(e)^2 \tag{6}$$

**Lemma 3.1.** *At round $t$, the LLM $M$ proposes a set of candidate operations $\mathcal{S}_t$. For any $\delta \in (0,1)$, with probability at least $1-\delta$, the deviation between the actual utility $g(e)$ and the predicted expected utility $\mu_t(e)$ is uniformly bounded for all $e \in \mathcal{S}_t$:*

$$\mathbb{P}\Big(\forall t \geq 1,\ \forall e \in \mathcal{S}_t:\ |g(e) - \mu_t(e)| \leq \sqrt{\beta_t}\, \sigma_t(e)\Big) \geq 1 - \delta, \quad \beta_t = 2\log\Big(\tfrac{|\mathcal{S}_t|\,\pi^2 t^2}{3\delta}\Big) \tag{7}$$

*Proof sketch.* Assuming the actual utility function $g(\cdot)$ can be linearly represented in the surrogate feature space $\phi(\cdot)$, the standardized error $(g(e) - \mu_t(e))/\sigma_t(e)$ is 1-sub-Gaussian, which gives the tail bound $\mathbb{P}(|g(e) - \mu_t(e)| > u\, \sigma_t(e)) \leq 2e^{-u^2/2}$. By applying a union bound over $\mathcal{S}_t$, we can then establish the confidence event in Equation. 7. The full proof is provided in Appendix A.1.

Given the predicted *expected utility* $\mu_t(e)$ and the *uncertainty* $\sigma_t^2(e)$ of each candidate operation $e$ at round $t$ of the feature engineering process, when human expertise is not available, the selection is solely based on the surrogate model's estimation. We adopt the Upper Confidence Bound (UCB) selection function (Auer et al., 2002; AV et al., 2022; Xu et al., 2024a) to balance exploitation of

---

[2]In this study, we instantiate $\phi_{\text{embedding}}$ with OpenAI's `text-embedding-3-small` model.

[3]In this study, we set $\mathcal{P}(\boldsymbol{\theta}) = \mathcal{N}(\boldsymbol{0}, I)$ as the prior over network parameters.

high predicted utility and exploration of uncertain operations. Specifically, for each round $t \leq T$ and LLM-proposed candidate operation $e \in \mathcal{S}_t$, the UCB selection function is defined as:

$$\text{UCB}_t(e) = \mu_t(e) + \sqrt{\beta_t}\,\sigma_t(e) \tag{8}$$

where $\beta_t = 2\log\left(\frac{|\mathcal{S}_t|\,\pi^2 t^2}{3\delta}\right)$ is set according to Lemma 3.1, which guarantees that, with probability at least $1 - \delta$ [4], $g(e) \in \left[\text{LCB}_t(e),\,\text{UCB}_t(e)\right]$ for all $e \in \mathcal{S}_t$, where $\text{LCB}_t(e) = \mu_t(e) - \sqrt{\beta_t}\,\sigma_t(e)$.

### 3.3 SELECTION OF LLM-PROPOSED FEATURE OPERATIONS WHEN HUMAN EXPERTISE IS AVAILABLE

When the human collaborator is available, we next proceed to explore how to selectively elicit and incorporate their preference feedback into the selection process of LLM-proposed operations.

***Selection of the Feature Operation Candidate Pair for Human Preference Feedback.*** Specifically, the human expertise in the feature engineering process is modeled as a stochastic oracle $\kappa$. At round $t$, given a pair of feature operations $(e_t^a, e_t^b)$, the oracle elicits a binary response $\kappa(e_t^a, e_t^b) = Z_t \in \{+1, -1\}$, where $Z_t = +1$ indicates $e_t^a \succ e_t^b$ and $Z_t = -1$ indicates $e_t^b \succ e_t^a$. We begin by describing how to select the pair $(e_t^a, e_t^b)$ to obtain human feedback at each round $t$. If the human collaborator $\kappa$ is not available at round $t$, we directly follow the UCB function (Equation 8) to select $e_t^a = \arg\max_{e \in \mathcal{S}_t} \text{UCB}_t(e)$ to evaluate in this round. When human expertise $\kappa$ is available, an additional operation $e_t^b$ can be selected from the remaining pool $S_t \setminus \{e_t^a\}$ such that the preference feedback $Z_t$ over the pair $(e_t^a, e_t^b)$ yields the highest expected utility gain relative to the surrogate model's current choice $e_t^a$. To define the expected information gain, let $e_t^\star$ denote the true optimal feature transformation at round $t$. The prior regret of the current selection is defined as $r_t = g(e_t^\star) - g(e_t^a)$, which measures the utility gap between the surrogate's choice $e_t^a$ and the unknown optimum $e_t^\star$. If we select another candidate $e_t^b$ and query the human oracle $\kappa$ to obtain preference feedback $Z_t$, we may revise the final decision to $e_t' \in \{e_t^a, e_t^b\}$ according to the feedback. The new regret becomes $r_t' = g(e_t^\star) - g(e_t')$. Therefore, the expected utility gain is defined as the expected reduction in regret resulting from incorporating human preference into the selection process:

$$U(e_t^a, e_t^b; \kappa) = \mathbb{E}_{Z_t}[r_t - r_t'] = \mathbb{E}_{Z_t}[g(e_t') - g(e_t^a)] \tag{9}$$

**Lemma 3.2.** *By the Lemma 3.1, let $e_t^a \in \mathcal{S}_t$ be the UCB choice, the following holds for any operation $e_t^b \in S_t \setminus \{e_t^a\}$ and $1 \leq t \leq T$:*

$$U(e_t^a, e_t^b; \kappa) = \mathbb{E}_{Z_t}[r_t - r_t'] \leq \max\{UCB_t(e_t^b) - LCB_t(e_t^a), 0\} \tag{10}$$

*Proof sketch.* Under the feature operation selection pair $(e_t^a, e_t^b)$, the human preference feedback can switch the round-$t$ choice of feature operation between $e_t^a$ and $e_t^b$, so $r_t - r_t' \leq \max\{g(e_t^b) - g(e_t^a), 0\}$. By Lemma 3.1, we have $g(e_t^b) \leq \text{UCB}_t(e_t^b)$ and $g(e_t^a) \geq \text{LCB}_t(e_t^a)$, which together gives the lemma. The full proof is provided in Appendix. A.2.

**Corollary 3.1.** *By Lemma 3.2, we have:*

$$U(e_t^a, e_t^b; \kappa) \leq \max\left\{UCB_t(e_t^b) - LCB_t(e_t^a), 0\right\} \leq \sqrt{\beta_t}\left(\sigma_t(e_t^a) + \sigma_t(e_t^b)\right) \tag{11}$$

Based on Lemma 3.2, the candidate operation $e_t^b$ is selected to be paired with $e_t^a$ for human preference feedback:

$$e_t^b = \text{argmax}_{e \in S_t \setminus \{e_t^a\}} \text{UCB}_t(e) \tag{12}$$

***Selective Elicitation for Human Preference Feedback.*** However, since eliciting human feedback inevitably incurs cognitive costs and increases the expert's workload, it is impractical to query the human collaborator in every round. Instead, feedback should be requested only when it is expected to yield gains that outweigh the associated costs, and incorporating human expertise can guide the selection algorithm toward more informed and effective decisions. Given the candidate pair $\{e_t^a, e_t^b\}$, we first consider whether human expert feedback has the potential to provide additional utility beyond the current selection. If $\text{UCB}_t(e_t^b) \leq \text{LCB}_t(e_t^a)$, then by Lemma 3.1, it indicates

---

[4]In this study, $\delta$ is set as 0.1 to control the confidence interval.

that $g(e_t^b) \leq g(e_t^a)$. In this case, the current feature operation $e_t^a$ strictly outperforms the candidate $e_t^b$, leaving no possibility for human expertise to improve the utility further. To prevent this, our first condition requires that the two confidence intervals should overlap to ensure that there remains uncertainty about which operation is better, thereby leaving the "room" for human expertise to potentially improve the selection:

$$(C1) \; Overlap: \quad \text{UCB}_t(e_t^b) > \text{LCB}_t(e_t^a) \tag{13}$$

Even when overlap exists, however, not all human feedback are equally valuable. The second consideration is whether the underlying improvement of utility is sufficiently large to justify the cognitive cost of involving the expert. Let $\gamma_\kappa$ denote the cost of eliciting human feedback in each round [5]. By Corollary 3.1, the maximum possible utility gain is upper-bounded by $\sqrt{\beta_t}(\sigma_t(e_t^a) + \sigma_t(e_t^b))$. To prevent unprofitable queries, we therefore impose the following requirement to guarantees the human feedback is only triggered when the potential utility improvement might outweigh the cost:

$$(C2) \; Uncertainty: \quad \sqrt{\beta_t}(\sigma_t(e_t^a) + \sigma_t(e_t^b)) \geq \gamma_\kappa \tag{14}$$

Taken together, the human feedback query is only elicited if and only if these two conditions hold:

$$(C1) \; Overlap: \quad \text{UCB}_t(e_t^b) > \text{LCB}_t(e_t^a) \quad \text{and} \quad (C2) \; Uncertainty: \quad \sqrt{\beta_t}(\sigma_t(e_t^a) + \sigma_t(e_t^b)) \geq \gamma\kappa \tag{15}$$

***Posterior Selection with Human Preference Feedback.*** Once the condition is satisfied and preference feedback $Z_t = \kappa(e_t^a, e_t^b)$ is elicited, the model distribution $q_t(\boldsymbol{\theta})$ for the surrogate model $\hat{g}$, which is constructed based on the past performance history $H_t$ using Equation 5, serves as the model's current belief about which feature operations may be useful. We then proceed to determine the final feature transformation for round $t$ by integrating this model belief $q_t(\boldsymbol{\theta})$ with the elicited human preference $Z_t$. Instead of making a direct selection based on $Z_t$, we treat $Z_t$ as a probabilistic observation that provides information about the relative utility between the two candidate operations. Specifically, the feedback likelihood is modeled using a probit function:

$$\mathcal{P}(Z_t \mid \boldsymbol{\theta}, e_t^a, e_t^b) = \Phi\Big(\eta Z_t\big[\hat{g}(\phi(e_t^a); \boldsymbol{\theta}) - \hat{g}(\phi(e_t^b); \boldsymbol{\theta})\big]\Big), \quad \text{where } \boldsymbol{\theta} \sim q_t(\boldsymbol{\theta}) \tag{16}$$

where $\Phi(\cdot)$ is the standard normal cumulative distribution function (CDF), and $\eta$ is a hyperparameter that controls the confidence level of feedback [6]. Assuming that the elicited human feedback $Z_t$, derived from human expertise, is conditionally independent of the performance history $H_t$, the posterior distribution $q_t'(\boldsymbol{\theta})$ can be updated as:

$$\text{KL}\big(q_t'(\boldsymbol{\theta}) \,\|\, \mathcal{P}(\boldsymbol{\theta} \mid H_t, Z_t, e_t^a, e_t^b)\big) = \int q_t'(\boldsymbol{\theta}) \log \frac{q_t'(\boldsymbol{\theta})}{\mathcal{P}(\boldsymbol{\theta} \mid H_t, Z_t, e_t^a, e_t^b)} \, d\boldsymbol{\theta}$$

$$= \int q_t'(\boldsymbol{\theta}) \left( \log \frac{q_t'(\boldsymbol{\theta})}{\mathcal{P}(\boldsymbol{\theta} \mid H_t)} - \log \mathcal{P}(Z_t \mid \boldsymbol{\theta}, e_t^a, e_t^b) + \log \mathcal{P}(Z_t \mid H_t, e_t^a, e_t^b) \right) d\boldsymbol{\theta} \tag{17}$$

$$\approx \text{KL}\big(q_t'(\boldsymbol{\theta}) \,\|\, q_t(\boldsymbol{\theta})\big) - \mathbb{E}_{q_t'(\boldsymbol{\theta})}\big[\log \mathcal{P}(Z_t \mid \boldsymbol{\theta}, e_t^a, e_t^b)\big] + \mathbb{E}_{q_t'(\boldsymbol{\theta})}\big[\log \mathcal{P}(Z_t \mid H_t, e_t^a, e_t^b)\big]$$

With the updated posterior distribution $q_t'(\boldsymbol{\theta})$, we can then make the selection of the final feature operation to be evaluated for this round $t$:

$$e_t^{\text{selected}} = \text{argmax}_{e \in \{e_t^a, e_t^b\}} \text{UCB}_t(e) \tag{18}$$

Finally, Algorithm 1 summarizes how does the proposed framework perform the iterative selection of LLM-proposed feature transformation operations in the feature engineering process.

## 4 EVALUATIONS

### 4.1 EXPERIMENTAL SETUP

***Dataset.*** Following previous work (Hollmann et al., 2023; Bordt et al., 2024), we select 18 datasets from Kaggle and UCI Irvine, which contain a mix of categorical and numerical features for classification or regression tasks. Since LLMs are trained on large-scale public data, which may include

---

[5]$\gamma_\kappa$ is empirically set to 4 in this study to balance the trade-off between final performance and efficiency for the elicitation of human feedback.

[6]$\eta$ is set as 1 in this study.

Table 1: Comparing the performance of the proposed method, LLM-based baselines, and non-LLM-based baselines on 13 classification datasets in terms of **AUROC (%)** with GPT-4o as the backbone model for all LLM-based methods, evaluated using MLP and XGBoost as the tabular learning models, respectively. The best method in each row is highlighted in blue, and the best baseline method is highlighted in light blue. The number in the brackets () indicate the error reduction rate compared to the best baseline method. All results are averaged over 5 runs.

| Dataset | MLP | | | | | | XGBoost | | | | | |
|---|---|---|---|---|---|---|---|---|---|---|---|---|
| | OpenFE | AutoGluon | CAAFE | OCTree | Ours (w/o human) | Ours (w/ human) | OpenFE | AutoGluon | CAAFE | OCTree | Ours (w/o human) | Ours (w/ human) |
| flight | 93.3 | 92.6 | 92.9 | 94.8 | 96.9 (+40.4%) | 97.3 (+48.1%) | 95.7 | 95.4 | 95.2 | 96.4 | 97.6 (+33.3%) | 98.0 (+44.4%) |
| wine | 77.2 | 77.2 | 77.6 | 78.2 | 78.5 (+1.4%) | 78.7 (+2.3%) | 81.3 | 81.0 | 80.9 | 82.1 | 82.9 (+4.5%) | 83.3 (+6.7%) |
| loan | 95.3 | 95.4 | 95.7 | 95.9 | 96.0 (+2.4%) | 96.1 (+4.9%) | 96.2 | 96.0 | 96.1 | 96.5 | 96.9 (+11.4%) | 97.1 (+17.1%) |
| diabetes | 81.1 | 82.4 | 82.8 | 82.8 | 83.0 (+1.2%) | 83.0 (+1.2%) | 84.1 | 83.9 | 83.9 | 84.4 | 85.2 (+5.1%) | 84.8 (+2.6%) |
| titanic | 84.1 | 84.3 | 86.3 | 86.5 | 86.8 (+2.2%) | 87.0 (+3.7%) | 85.0 | 84.8 | 87.0 | 87.4 | 87.9 (+4.0%) | 88.3 (+7.1%) |
| travel | 80.4 | 80.3 | 81.1 | 81.7 | 82.0 (+1.6%) | 82.3 (+3.3%) | 83.6 | 83.2 | 83.6 | 84.6 | 85.3 (+4.5%) | 85.7 (+7.1%) |
| ai_usage | 67.8 | 67.5 | 68.2 | 68.0 | 68.5 (+0.9%) | 68.3 (+0.3%) | 71.8 | 71.5 | 71.3 | 72.4 | 73.3 (+3.3%) | 73.8 (+5.1%) |
| water | 53.7 | 53.2 | 56.7 | 57.9 | 58.7 (+1.9%) | 59.3 (+3.3%) | 56.7 | 56.1 | 59.8 | 61.7 | 63.2 (+3.9%) | 64.1 (+6.3%) |
| heart | 92.2 | 92.3 | 92.6 | 93.1 | 93.4 (+4.3%) | 93.6 (+7.2%) | 93.6 | 93.5 | 93.6 | 94.3 | 95.1 (+14.0%) | 94.8 (+8.8%) |
| adult | 90.5 | 90.4 | 90.8 | 90.9 | 91.3 (+4.4%) | 91.4 (+5.5%) | 91.6 | 91.3 | 91.5 | 92.0 | 92.4 (+5.0%) | 92.8 (+10.0%) |
| customer | 84.6 | 84.5 | 84.9 | 84.8 | 85.1 (+1.3%) | 85.1 (+1.3%) | 85.3 | 85.0 | 85.3 | 85.2 | 85.8 (+3.4%) | 86.3 (+6.8%) |
| personality | 94.4 | 94.1 | 95.0 | 95.4 | 96.1 (+15.2%) | 96.1 (+15.2%) | 96.4 | 96.2 | 96.6 | 97.1 | 97.4 (+10.3%) | 97.6 (+17.2%) |
| *conversion* | 90.7 | 90.6 | 90.9 | 91.1 | 92.6 (+16.9%) | 92.9 (+20.2%) | 91.2 | 91.9 | 92.1 | 92.4 | 93.5 (+5.7%) | 93.9 (+11.5%) |

Table 2: Comparing the performance of different LLMs as backbones for LLM-based methods in terms of average **AUROC (%)** on 13 classification datasets, evaluated using MLP and XGBoost as the downstream tabular learning models, respectively. All results are averaged over 5 runs.

| Model | MLP | | | | | | XGBoost | | | | | |
|---|---|---|---|---|---|---|---|---|---|---|---|---|
| | OpenFE | AutoGluon | CAAFE | OCTree | Ours (w/o) | Ours (w/ human) | OpenFE | AutoGluon | CAAFE | OCTree | Ours (w/o) | Ours (w/ human) |
| Deepseek-v3 | 83.5 | 83.5 | 84.9 | 85.5 | 86.1 | 86.4 | 85.6 | 85.4 | 86.6 | 87.3 | 88.2 | 88.6 |
| GPT-3.5-turbo | 83.5 | 83.5 | 83.2 | 84.2 | 84.6 | 85.1 | 85.6 | 85.4 | 85.2 | 86.0 | 86.5 | 87.1 |
| GPT-4o | 83.5 | 83.5 | 84.3 | 84.7 | 85.3 | 85.5 | 85.6 | 85.4 | 85.9 | 86.7 | 87.4 | 87.4 |
| GPT-5 | 83.5 | 83.5 | 85.5 | 85.8 | 85.9 | 86.5 | 85.6 | 85.4 | 87.1 | 87.7 | 88.0 | 88.7 |

information on how to construct successful features for these widely used datasets, we additionally include a proprietary company dataset of predicting users' conversion intentions, which cannot be accessed by the LLM, to provide a more robust evaluation. Detailed information about each dataset is provided in Appendix C.1.

***Baselines and Operationalizing the Proposed Method.*** We consider both AutoML and LLM-based feature engineering methods as baselines. For AutoML methods, we include OpenFE (Zhang et al., 2023) and AutoGluon (Erickson et al., 2020). For LLM-based methods, we consider CAAFE(Hollmann et al., 2023) and OCTree(Nam et al., 2024a). To evaluate the effectiveness of feature engineering, we employ MLP (Rumelhart et al., 1986) and XGBoost (Chen & Guestrin, 2016) as downstream prediction models. For non-LLM methods, the feature engineering process proceeds until convergence to their best performance. For all LLM-based methods, we use GPT-4o (OpenAI, 2024) as the backbone model to generate feature operations in each round with a sampling temperature of 1, and the maximum iteration budget is set at 50. For the proposed method, the LLM generates 15 candidate feature transformation operations per prompt in each iteration. We specifically evaluate our method under two settings: one where the human collaborator is absent (*w/o Human*), and one where the human collaborator is available (*w/ Human*). To simulate the *w/ Human* setting, we employ GPT-4o as a proxy as the human expert to provide the preference feedback. In particular, for each dataset, we first fit a base classifier and use SHAP (Lundberg & Lee, 2017) to identify the most important features for the task. These feature importance scores are then converted into the expert prompts, enabling the proxy model to judge which feature pairs are more promising. Each dataset is randomly partitioned into 80% training and 20% validation sets, and this process is repeated for 5 iterations to evaluate each method's performance. For detailed prompt template used in each round, please see Appendix D.

## 4.2 EVALUATION RESULTS

Table 1 presents the comparison of final feature engineering performance across 13 classification datasets using the proposed method, LLM-based baselines, and AutoML baselines with GPT-4o model as the backbone to generate feature operations for all LLM-based methods. Overall, we observe that both *Ours (w/o human)* and *Ours (w/ human)* consistently outperform the best baseline methods across different datasets. Specifically, when using MLP as the evaluation model, Ours (w/o human) achieves an average error rate reduction of 7.24%, while Ours (w/ human) achieves 8.96% compared to the best baseline. Similarly, when using the XGBoost as the evaluator, *Ours (w/o human)* achieves 9.02%, and *Ours (w/ human)* achieves 11.23% average error rate reduction over the best baselines. Below, we summarized key observations.

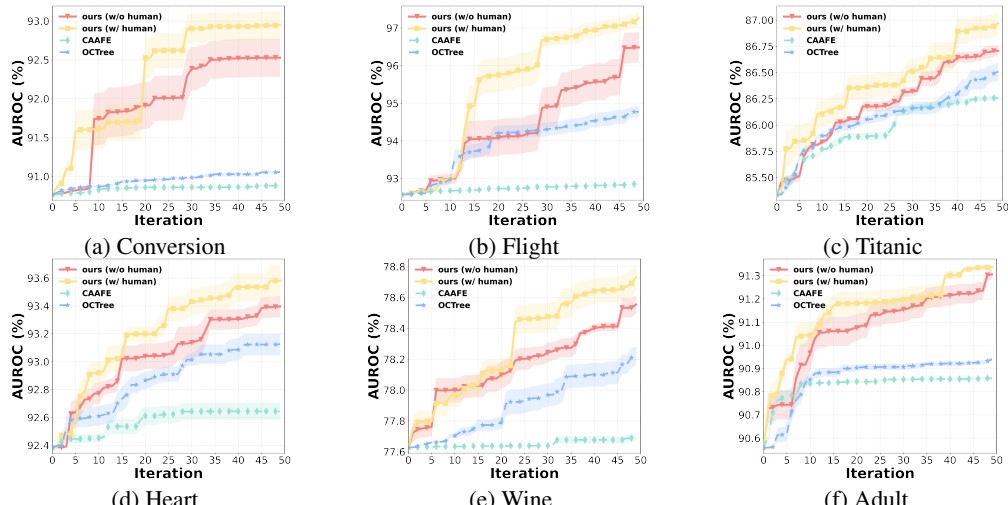

Figure 1: Comparing the performance trajectories of the proposed method with two LLM-based baselines (CAAFE and OCTree) in the feature engineering process, using an iteration budget of 50 and MLP as the tabular learner across six tasks. Error shade indicates the standard error of the mean.

**LLM-based feature engineering methods outperform traditional non-LLM automatic feature engineering approaches.** We found that feature engineering methods leveraging large language models (e.g., CAAFE, OCTree, and our proposed method) generally outperform traditional automatic feature engineering methods such as OpenFE and AutoGluon. We attribute this improvement to the strong semantic understanding, reasoning, and generative capabilities of LLMs, which enable them to efficiently generate effective feature transformation operations tailored to different tasks.

**Explicitly modeling utility and uncertainty of feature operations improves LLM-powered feature engineering.** Unlike prior LLM-based methods such as CAAFE and OCTree, our proposed approach generally leads to improved performance across different tasks. As a proof of concept, on the proprietary company-owned *conversion dataset*, OCTree achieves an AUROC of 91.1%. In contrast, *Ours (w/o human)* achieves 92.4% and *Ours (w/ human)* achieves 92.6% further under the same iteration budget as OCTree.

**Incorporating human preference feedback improves performance.** Finally, we evaluated the impact of incorporating human preference feedback on the performance of our proposed feature engineering method. We observed that the addition of human feedback consistently improves the method's performance across most tasks compared to when no feedback is available. Specifically, in the MLP model, the incorporation of human feedback increases the average error reduction rate by 1.72% compared to the best baseline. Similarly, in the XGBoost model, the improvement increases further to 3.21% of the average error reduction. For the remaining regression tasks, we observe similar performance trends across both baseline methods and our proposed method (see Appendix C.2).

In addition to using GPT-4o as the backbone model, we further compare the performance of all LLM-based feature engineering methods under alternative backbone generators. For the *Ours (w/ human)*, we continue to use GPT-4o to simulate human preference feedback to ensure consistency across different backbone settings. As shown in Table 2, we evaluate four LLMs—DeepSeek-V3 (Liu et al., 2024), GPT-3.5-Turbo (Ouyang et al., 2022), GPT-4o (OpenAI, 2024), and GPT-5 (OpenAI, 2025) across the same 13 datasets using both MLP and XGBoost as downstream tabular models. Across all backbones, both *Ours (w/o human)* and *Ours (w/ human)* consistently outperform LLM-based and non-LLM baselines, demonstrating that the proposed framework is robust to the choice of backbone model. Even with a weaker generator such as GPT-3.5-Turbo, our method can still maintain strong performance.

## 4.3 ANALYSIS OF ITERATIVE GAINS IN FEATURE ENGINEERING

To better understand the advantage of our approach over LLM-based baselines that are based solely on LLM's internal heuristics to generate and select feature operations , we analyze iterative gains throughout the feature engineering process by comparing the performance trajectories of our method (*w/o human* and *w/ human*) against LLM-based baselines. Figures 1a to Figure 1f present the performance trajectories of our method versus CAAFE and OCTree during the feature engineering

Table 3: Runtime breakdown of the proposed feature engineering pipelines when varying the number of initial features from 10 to 10,000 with a full dataset of 10,000 instances.

| Features | LLM (s) | Surrogate (s) | UCB (s) | Eval (s) |
|---|---|---|---|---|
| 10 | 1.82 | 0.17 | 0.006 | 1.79 |
| 50 | 1.82 | 0.16 | 0.005 | 1.24 |
| 100 | 1.82 | 0.19 | 0.005 | 1.78 |
| 1,000 | 1.82 | 0.20 | 0.009 | 8.4 |
| 10,000 | 1.82 | 0.57 | 0.018 | 23.4 |

Table 4: Runtime breakdown of the proposed feature engineering pipeline when varying the number of training instances from 1,000 to 100,000 with 100 initial feature columns.

| Samples | LLM (s) | Surrogate (s) | UCB (s) | Eval (s) |
|---|---|---|---|---|
| 1,000 | 1.82 | 0.17 | 0.005 | 0.28 |
| 5,000 | 1.82 | 0.18 | 0.005 | 0.89 |
| 10,000 | 1.82 | 0.23 | 0.006 | 1.47 |
| 50,000 | 1.82 | 0.18 | 0.006 | 5.22 |
| 100,000 | 1.82 | 0.18 | 0.005 | 10.65 |

process, under an iteration budget of 50, using MLP as the tabular learner. Visually, we observe that unlike CAAFE and OCTree, which often become trapped in local optima or experience performance stagnation, our method is able to identify high-impact feature operations that lead to notable performance jumps at various points and make steady progress during the iteration process. Furthermore, when investigating the impact of incorporating the selectively elicited human feedback, we observe that it often helps the algorithm redirect the feature search toward more promising operations, which enables the algorithm to make sharper gains and achieve even higher performance compared to the setting without human input. Similar trends are observed in other datasets (see Appendix C.3).

## 4.4 ANALYSIS OF COMPUTATIONAL SCALABILITY OF THE PROPOSED METHOD

To understand the computational scalability of the proposed method, we measured the runtime of each component of the LLM-powered feature engineering pipeline within a single iteration, which consists of four steps:

1. calling the LLM to generate candidate feature transformations.
2. fitting the BNN surrogate model using the accumulated observations.
3. computing UCB scores for the proposed candidates.
4. evaluating the selected transformation using the downstream tabular model.

Here, steps 2 and 3 are the specific computational components of our framework, while steps 1 and 4 are shared across all LLM-based pipelines. We then conducted controlled experiments on synthetic binary-classification datasets by systematically varying the number of initial feature columns and the number of data instances to directly measure how each component of the pipeline scales under datasets of different sizes. All feature columns were sampled from standard normal distributions, labels from a Bernoulli distribution, and an MLP was used as the downstream evaluator. All LLM calls were made using GPT-4o, and for the downstream evaluation step, we fixed the MLP training to a single epoch to ensure consistent and comparable timing across all conditions. We first fixed the dataset size to 10,000 instances and varied the number of initial features from 10 to 10,000. As shown in Table 3, the time required for fitting surrogate model and UCB score computation increases only mildly with feature dimensionality, whereas the downstream evaluator dominates the runtime as the number of initial features grows. For instance, with 10,000 initial features, the surrogate model and UCB steps together account for only about 2.2% of the total runtime. Next, we fixed the number of initial features to 100 and varied the dataset size from 1,000 to 100,000 rows. As shown in Table 4, the surrogate fitting and UCB computation times remain nearly constant across all sample sizes and contribute only a small percentage of the total runtime, because both operate at the feature-operation level and are independent of the number of data instances. In contrast, the downstream evaluation time increases with dataset size, as it requires training the MLP on the full dataset. Overall, these results indicate that our method scales favorably with both feature dimensionality and dataset size.

## 4.5 USER STUDY: HOW DO HUMANS PERFORM AND PERCEIVE WITH OUR METHOD?

Finally, to understand how actual users would collaborate with our algorithm to complete the feature engineering task, we conducted a user study, where we selected the flight dataset of predicting passengers' satisfaction levels as the feature engineering task. We designed three treatments by varying the ways in which participants could collaborate with the LLM to complete the task:

- CONTROL: The human fully leads the feature engineering process. In each round, the human needs to provide language instructions about what to do, and the LLM will parse the instructions to create the feature operations.
- SELF REASONING (SR): In this treatment, the overall algorithm remains the same as our proposed method, except that the querying mechanism is modified so the LLM autonomously de-

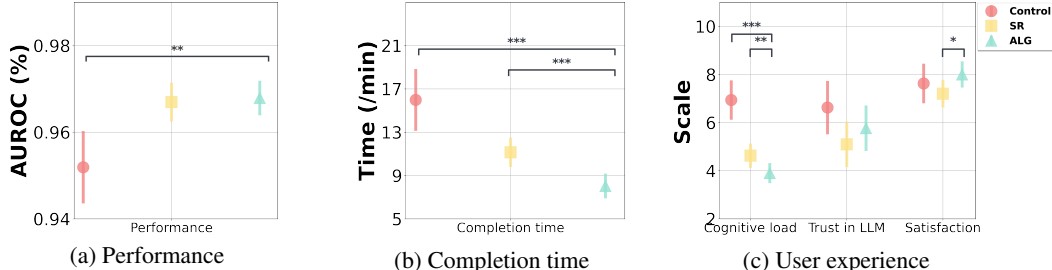

| (a) Performance | (b) Completion time | (c) User experience |

Figure 2: Comparing participants' *average* final feature engineering performance, completion time, and the user experience perceptions for the flight satisfaction prediction task in the CONTROL, SR, and our ALG treatment, respectively. Error bars represent the 95% confidence intervals of the mean values. *, **, and *** denote statistical significance levels of 0.1, 0.05, and 0.01 respectively.

- cides in each round whether to query human preference feedback, rather than being triggered by our algorithm.
- OURS (ALG): In this treatment, the human preference feedback is triggered by our proposed method.

We recruited 31 machine learning engineers/researchers or graduate students with a background in AI/ML as participants[7]. Each participant was randomly assigned to one of the three treatments. At the beginning of the study, participants were provided with SHAP-generated explanations, which highlighted the importance of each original feature in the dataset, to equip them with task-specific expertise. Following this, participants proceeded with the feature engineering task in different ways of collaborating with the LLM depending on the assigned treatment until the iteration budget was exhausted. We used GPT-4o as the backbone model to generate feature transformation operations and MLP as the tabular learner to evaluate the utility of the feature operations, with the iteration budget of feature engineering process set to 40 for all treatments. Finally, participants were required to complete an exit survey to report their perceptions on the overall feature engineering process. In this survey, we used the NASA Task Load Index (Hart & Staveland, 1988) to measure the cognitive load experienced by participants during the feature engineering process, including their perceived mental demand, time pressure, effort level, and frustration. In addition, participants were also asked to rate the overall satisfaction, and the trust in using the system for future on a scale from 1 to 10.

Figure 2a compares the participants' average final feature engineering performance across three treatments. One-way ANOVA and subsequent Tukey's HSD test show that participants who worked under our ALG framework demonstrated significantly higher final feature engineering task performance compared to those in the CONTROL treatment ($p = 0.011$). Figure 2b compares participants' average task completion time across the three treatments. Visually, we observe that participants in our ALG framework exhibited higher time efficiency compared to the other two conditions. One-way ANOVA and further Tukey's HSD test show that participants under our ALG framework had significantly lower completion times compared to those in CONTROL ($p < 0.001$) and those in SR ($p = 0.02$). We finally move on to investigate how participants perceive their experience under different treatments. As shown in Figure 2c, participants in our ALG framework reported significantly lower cognitive load compared to those in CONTROL ($p < 0.001$) and those in SR ($p = 0.043$), as well as a marginally significantly higher satisfaction compared to those in SR ($p = 0.072$).

## 5  CONCLUSION

In this paper, we propose a human–LLM collaborative feature engineering framework for tabular learning. We begin by decoupling the transformation operation proposal and selection processes, where LLMs are used solely to generate operation candidates, while the selection is guided by explicitly modeling the utility and uncertainty of each proposed operation. We then design a mechanism within the framework that selectively elicits and incorporates human expert preference feedback into the selection process to help identify more effective operations to explore. Our evaluations on the both synthetic study and real user study demonstrate that the proposed framework improves feature engineering performance across a variety of tabular datasets and reduces users' cognitive load during the feature engineering process.

---

[7]This study was approved by the author's Institutional Review Board.

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

# A   PROOF FOR TECHNICAL LEMMAS IN SECTION 3

## A.1   COMPLETE PROOF FOR LEMMA 3.1

Given the surrogate model posterior distribution $q_t(\boldsymbol{\theta}) = \mathcal{N}(\boldsymbol{\theta}; \boldsymbol{M}_t, \boldsymbol{\Sigma}_t)$, and the surrogate model $\hat{g}(\phi(e); \boldsymbol{\theta})$, we first perform a first-order Taylor expansion of $\hat{g}(\phi(e); \boldsymbol{\theta})$ around $\boldsymbol{M}_t$:

$$\hat{g}(\phi(e); \boldsymbol{\theta}) = \underbrace{\hat{g}(\phi(e); \boldsymbol{M}_t)}_{=: \, c_t(e)} + \underbrace{\nabla_{\boldsymbol{\theta}} \hat{g}^\top(\phi(e); \boldsymbol{M}_t)}_{=: \, \phi_t(e)^\top} \underbrace{(\boldsymbol{\theta} - \boldsymbol{M}_t)}_{=: \, \boldsymbol{w}} + R_2(e; \boldsymbol{\theta})$$

$$= c_t(e) + \phi_t(e)^\top \boldsymbol{w} + R_2(e; \boldsymbol{\theta})$$

where $\boldsymbol{w} \sim \mathcal{N}(\boldsymbol{w}; 0, \boldsymbol{\Sigma}_t)$. Since our surrogate model $\hat{g}$ is implemented as a multilayer percep­tron (MLP) with 1-Lipschitz activations (e.g., ReLU), based on the prior research (Boyd & Van­denberghe, 2004), this indicates that the second-order remainder $R_2(e; \boldsymbol{\theta})$ is locally bounded as $|R_2(e; \boldsymbol{\theta})| \leq C \|\boldsymbol{w}\|_2^2$ where $C$ is model architecture-dependent constant. We therefore omit $R_2$ and use the *linearized surrogate model* in the subsequent proof:

$$\hat{g}_{\text{lin}}(\phi(e); \boldsymbol{w}) = c_t(e) + \phi_t(e)^\top \boldsymbol{w}, \quad \text{where} \quad \boldsymbol{w} \sim \mathcal{N}(\boldsymbol{w}; 0, \boldsymbol{\Sigma}_t)$$

**Assumption A.1** (Utility Linearity). *Given some model weights $\boldsymbol{w}^\star \sim \mathcal{N}(\mathbf{0}, \boldsymbol{\Sigma}_t)$, the true utility of a feature transformation operation $g(e)$ can be linearly represented in the feature space $\phi(\cdot)$, i.e., $g(e) = c_t(e) + \phi_t(e)^\top \boldsymbol{w}^\star$.*

**Lemma A.1.** *By Assumption A.1, the standardized deviation $\Psi = \frac{g(e) - \mu_t(e)}{\sigma_t(e)}$ is 1-sub-Gaussian, i.e., $\mathbb{E}[\exp(\lambda \Psi)] \leq \exp\left(\frac{\lambda^2}{2}\right), \, \forall \lambda \in \mathbb{R}$.*

*Proof.* Given the predicted *expected utility* $\mu_t(e)$ of a candidate operation $e$ and its corresponding *uncertainty* $\sigma_t^2(e)$:

$$\mu_t(e) = \mathbb{E}_{q_t(\boldsymbol{\theta}))}[\hat{g}(\phi(e); \boldsymbol{\theta})], \quad \sigma_t^2(e) = \mathbb{E}_{q_t(\boldsymbol{\theta}))}[\hat{g}(\phi(e); \boldsymbol{\theta})^2] - \mu_t(e)^2$$

As the surrogate model $\hat{g}(\cdot)$ can be approximated as the linearized surrogate model $\hat{g}_{\text{lin}}(\cdot)$, the ex­pected utility $\mu_t(e)$ and the uncertainty $\sigma_t^2(e)$ can be represented as:

$$\mu_t(e) = \mathbb{E}[c_t(e) + \phi_t(e)^\top \boldsymbol{w}] = c_t(e), \quad \sigma_t^2(e) = \text{Var}(c_t(e) + \phi_t(e)^\top \boldsymbol{w}) = \phi_t(e)^\top \boldsymbol{\Sigma}_t \phi_t(e)$$

Consequently, the standard deviation $\Psi$ is:

$$\Psi = \frac{g(e) - \mu_t(e)}{\sigma_t(e)} = \frac{c_t(e) + \phi_t(e)^\top \boldsymbol{w}^\star - c_t(e)}{\sqrt{\phi_t(e)^\top \boldsymbol{\Sigma}_t \phi_t(e)}} = \frac{\phi_t(e)^\top \boldsymbol{w}^\star}{\sqrt{\phi_t(e)^\top \boldsymbol{\Sigma}_t \phi_t(e)}}$$

Since $\boldsymbol{w}^\star \sim \mathcal{N}(0, \boldsymbol{\Sigma}_t)$, we have $\phi_t(e)^\top \boldsymbol{w}^\star \sim \mathcal{N}(0, \phi_t(e)^\top \boldsymbol{\Sigma}_t \phi_t(e))$. Thus, $\Psi = \frac{\phi_t(e)^\top \boldsymbol{w}^\star}{\sqrt{\phi_t(e)^\top \boldsymbol{\Sigma}_t \phi_t(e)}} \sim \mathcal{N}(0, 1)$, and $\mathbb{E}[\exp(\lambda \Psi)] \leq \exp\left(\frac{\lambda^2}{2}\right)$.

**Lemma 3.1** *At round $t$, the LLM $M$ proposes a set of candidate operations $\mathcal{S}_t$. For any $\delta \in (0, 1)$, with probability at least $1 - \delta$, the deviation between the actual utility $g(e)$ and the predicted expected utility $\mu_t(e)$ is uniformly bounded for all $e \in \mathcal{S}_t$:*

$$\mathbb{P}\left(\forall t \geq 1, \, \forall e \in \mathcal{S}_t : \, |g(e) - \mu_t(e)| \leq \sqrt{\beta_t} \, \sigma_t(e)\right) \geq 1 - \delta, \quad \beta_t = 2 \log\left(\frac{|\mathcal{S}_t| \pi^2 t^2}{3\delta}\right)$$

*Proof.* By Lemma A.1, for any $u > 0$, we have:

$$\mathbb{P}(|g(e) - \mu_t(e)| > u\sigma_t(e)) = \mathbb{P}(|\Psi| > u) = 2\Phi(-u) \leq 2e^{-u^2/2}$$

where $\Phi$ is the standard normal CDF. At round $t$, given the LLM-proposed feature set $\mathcal{S}_t$, setting $u = \sqrt{\beta_t}$ and applying the union bound yields:

$$\mathbb{P}\left(\exists e \in \mathcal{S}_t : \, |g(e) - \mu_t(e)| > \sqrt{\beta_t} \, \sigma_t(e)\right) \leq \sum_{e \in \mathcal{S}_t} 2 \, e^{-\beta_t/2} = 2 |\mathcal{S}_t| \, e^{-\beta_t/2}$$

Let

$$\beta_t \;=\; 2\log\!\left(\frac{|\mathcal{S}_t|\,\pi^2 t^2}{3\delta}\right)$$

Then

$$\mathbb{P}\Big(\exists e \in \mathcal{S}_t:\; |g(e) - \mu_t(e)| > \sqrt{\beta_t}\,\sigma_t(e)\Big) \;\leq\; \frac{6\delta}{\pi^2 t^2}$$

Define the failure event at round $t$:

$$\mathcal{F}_t \;:=\; \Big\{\exists e \in \mathcal{S}_t:\; |g(e) - \mu_t(e)| > \sqrt{\beta_t}\,\sigma_t(e)\Big\}$$

Apply the union bound over all rounds $t \geq 1$, we have:

$$\mathbb{P}\!\left(\bigcup_{t=1}^{\infty}\mathcal{F}_t\right) \;\leq\; \sum_{t=1}^{\infty}\mathbb{P}(\mathcal{F}_t) \;\leq\; \sum_{t=1}^{\infty}\frac{6\delta}{\pi^2 t^2} \;=\; \delta$$

Thus, with probability at least $1 - \delta$, the confidence bound holds uniformly for all $e \in \mathcal{S}_t$, $\forall t \geq 1$.

### A.2 Complete proof for Lemma 3.2

**Lemma 3.2** *By the Lemma 3.1, let $e_t^a \in \mathcal{S}_t$ be the UCB choice, the following holds for any operation $e_t^b \in S_t \setminus \{e_t^a\}$ and $1 \leq t \leq T$:*

$$U(e_t^a, e_t^b; \kappa) \;=\; \mathbb{E}_{Z_t}\big[r_t' - r_t\big] \;\leq\; \max\{\mathrm{UCB}_t(e_t^b) - \mathrm{LCB}_t(e_t^a), 0\}$$

*Proof.* After receiving the human preference feedback $Z_t$, the posterior feature transformation operation is selected from the pair $\{e_t^a, e_t^b\}$. The regret reduction is defined as:

$$r_t - r_t' \;=\; \big[g(e_t^\star) - g(e_t^a)\big] - \big[g(e_t^\star) - g(e_t')\big] \;=\; g(e_t') - g(e_t^a),$$

where $e_t^\star$ is the optimal operation and $e_t' \in \{e_t^a, e_t^b\}$ is the final selected one. Since $e_t' \in \{e_t^a, e_t^b\}$, the regret reduction is bounded by:

$$r_t - r_t' \;\leq\; \max\big\{g(e_t^b) - g(e_t^a),\, 0\big\}.$$

Taking expectation over the preference feedback $Z_t$, we obtain:

$$U(e_t^a, e_t^b; \kappa) \;:=\; \mathbb{E}_{Z_t}\big[r_t - r_t'\big] \;\leq\; \max\big\{g(e_t^b) - g(e_t^a),\, 0\big\}.$$

Under the confidence event in Lemma 3.1, for any $e \in \mathcal{S}_t$, the true utility is bounded as:

$$g(e) \in \big[\mu_t(e) - \sqrt{\beta_t}\,\sigma_t(e),\; \mu_t(e) + \sqrt{\beta_t}\,\sigma_t(e)\big],$$

i.e.,

$$g(e) \leq \mathrm{UCB}_t(e) := \mu_t(e) + \sqrt{\beta_t}\,\sigma_t(e), \quad g(e) \geq \mathrm{LCB}_t(e) := \mu_t(e) - \sqrt{\beta_t}\,\sigma_t(e).$$

Therefore,

$$g(e_t^b) - g(e_t^a) \;\leq\; \mathrm{UCB}_t(e_t^b) - \mathrm{LCB}_t(e_t^a),$$

and thus,

$$U(e_t^a, e_t^b; \kappa) \;\leq\; \max\big\{\mathrm{UCB}_t(e_t^b) - \mathrm{LCB}_t(e_t^a),\, 0\big\}.$$

## B Algorithm

Algorithm 1 summarizes how does the proposed framework perform the iterative selection of LLM-proposed feature transformation operations in the feature engineering process.

## C Evaluations (Additional Details)

### C.1 Descriptions of the datasets used in the main study

We evaluate our methods across 13 widely-used Kaggle classification datasets, covering diverse domains such as healthcare, finance, customer behavior, and public records. A brief description of each dataset is provided below:

---

**Algorithm 1** Iterative Selection of LLM-Proposed Feature Transformation Operations

---

**Input:** Training Dataset $\mathcal{D}_{\text{train}}$, validation set $\mathcal{D}_{\text{eval}}$, LLM $M$, a tabular learner $f$, Iteration Budget $T$

1: Initialize $H_1 \leftarrow \varnothing$, the feature operation pool $S_0 \leftarrow \varnothing$
2: **for** $t = 1$ to $T$ **do**
3:     $\mathcal{S}_t \leftarrow \{\text{feature operations proposed by } M \text{ in the round } t\} \cup \mathcal{S}_{t-1} \setminus \{e_{t-1}^{\text{selected}}\}$
4:     Fit the surrogate model $q_t(\boldsymbol{\theta})$ via equation 5
5:     Select $e_t^a$ via equation 8
6:     **if** human expertise $\kappa$ is not available **then**
7:         $e_t^{\text{selected}} \leftarrow e_t^a$
8:     **else**
9:         Select $e_t^b$ via equation 12
10:         **if** the trigger conditions (equation 15) all hold **then**
11:             Query human preference feedback: $Z_t \leftarrow \kappa(e_t^a, e_t^b)$
12:             Update the surrogate model $q_t'(\boldsymbol{\theta})$ via equation 17
13:             $e_t^{\text{selected}} \leftarrow \arg\max_{e \in \{e_t^a, e_t^b\}} \text{UCB}_t(e)$ via equation 18
14:         **else**
15:             $e_t^{\text{selected}} \leftarrow e_t^a$
16:     Fit the tabular learner $f$ on $\mathcal{D}_{\text{train}} \oplus e_t^{\text{selected}}$, and evaluate $g(e_t^{\text{selected}})$ on $\mathcal{D}_{\text{val}} \oplus e_t^{\text{selected}}$
17:     $H_{t+1} \leftarrow H_t \cup \{(e_t^{\text{selected}}, g(e_t^{\text{selected}}))\}$
18:     **if** $g(e_t^{\text{selected}}) > 0$ **then**
19:         $\mathcal{D}_{\text{train}} \leftarrow \mathcal{D}_{\text{train}} \oplus e_t^{\text{selected}}$, $\mathcal{D}_{\text{eval}} \leftarrow \mathcal{D}_{\text{eval}} \oplus e_t^{\text{selected}}$
20: **return** $\{e_t^{\text{selected}}\}_{t=1}^T$, $\mathcal{D}_{\text{train}}$, and $\mathcal{D}_{\text{eval}}$

---

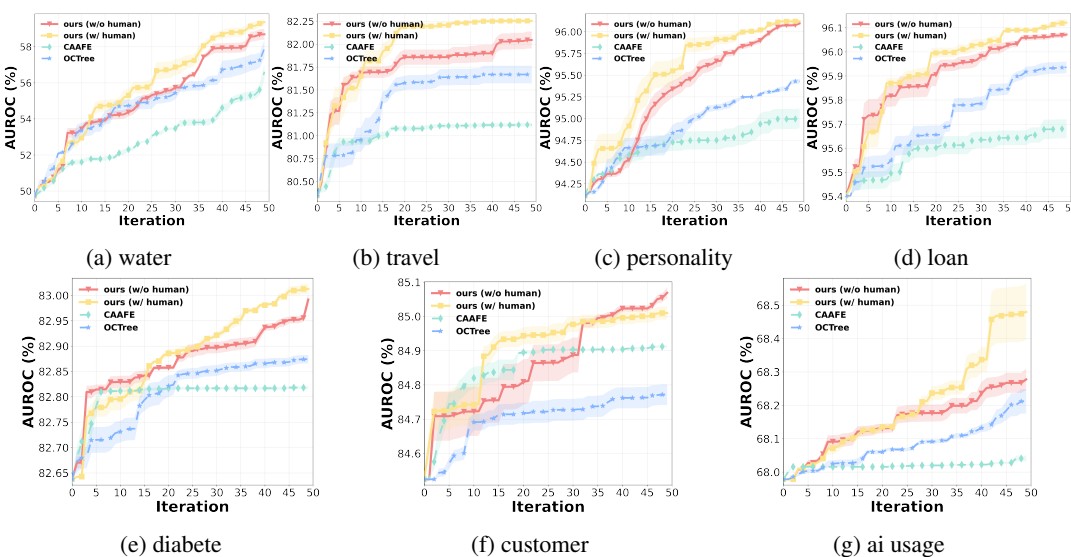

Figure C.1: Comparing the performance trajectories of the proposed method with two LLM-based baselines (CAAFE and OCTree) in the feature engineering process, using an iteration budget of 50 and MLP as the tabular learner across seven tasks. Error shade indicates the standard error of the mean.

## C.2 EVALUATION RESULTS (ADDITIONAL DETAILS)

Table C.2 compares the performance of the proposed method, LLM-based baselines, and non-LLM-based baselines on 5 regression datasets in terms of error reduction rate (%) compared with the base learner with GPT-4o as the backbone model for all LLM-based methods, evaluated using MLP and XGBoost as the tabular learning models, respectively. We again observed that our proposed methods can outperform the baselines including AutoML methods and LLM-powered feature engineering approaches.

| Dataset | Description | #Features | #Instances | Task Type |
|---------|-------------|-----------|------------|-----------|
| flight | Predict whether a flight is delayed based on schedule and airline attributes. | 22 | 25,976 | Classification |
| wine | Classify wine quality using physicochemical test results. | 11 | 945 | Classification |
| loan | Predict loan approval based on applicant demographic and financial attributes. | 13 | 45,000 | Classification |
| Diabetes | Diagnose diabetes from medical measurements of female patients. | 21 | 40,000 | Classification |
| titanic | Predict passenger survival on the Titanic from demographic and ticket info. | 8 | 891 | Classification |
| travel | Predict whether a customer purchased travel insurance or filed a claim. | 8 | 63,326 | Classification |
| ai_usage | Predict whether a survey respondent reports using AI tools. | 8 | 10,000 | Classification |
| water | Classify whether water is potable given physicochemical properties. | 9 | 3,276 | Classification |
| heart | Diagnose presence of heart disease based on clinical measurements. | 11 | 918 | Classification |
| adult | Predict if income exceeds $50K based on census demographic data. | 14 | 32,561 | Classification |
| customer | Predict whether a telecom customer will churn from usage statistics. | 20 | 7,043 | Classification |
| personality | Predict Big Five personality types from survey responses. | 7 | 2,900 | Classification |
| conversion | Predict whether an online shopper will convert (make a purchase). | 178 | 15,000 | Classification |
| housing | Predict house price based on the information of the house. | 9 | 20640 | Regression |
| forest | Predict burned area in forest fires based on geographic information. | 12 | 517 | Regression |
| bike | Predict daily bike rental counts from weather and calendar info. | 9 | 17,414 | Regression |
| crab | Predict age of crabs based on biometric measurements. | 8 | 3,893 | Regression |
| insurance | Predict the insurance cost. | 6 | 1,339 | Regression |

Table C.1: Summary of the datasets used in our experiments.

Table C.2: Comparing the performance of the proposed method, LLM-based baselines, and non-LLM-based baselines on 5 regression datasets in terms of normalized root mean square error with GPT-4o as the backbone model for all LLM-based methods, evaluated using MLP and XGBoost as the tabular learning models, respectively. The best method in each row is highlighted in blue, and the best baseline method is highlighted in light blue. The number in the brackets () indicate the error reduction rate compared to the best baseline method. All results are averaged over 5 runs.

| Dataset | MLP | | | | | | XGBoost | | | | | |
|---------|--------|-----------|-------|--------|------------------|-----------------|--------|-----------|-------|--------|------------------|-----------------|
| | OpenFE | AutoGluon | CAAFE | OCTree | Ours (w/o human) | Ours (w/ human) | OpenFE | AutoGluon | CAAFE | OCTree | Ours (w/o human) | Ours (w/ human) |
| housing | 0.316 | 0.319 | 0.292 | 0.283 | 0.270 | 0.266 | 0.228 | 0.231 | 0.224 | 0.221 | 0.216 | 0.214 |
| forest | 1.851 | 1.851 | 1.750 | 1.724 | 1.655 | 1.621 | 1.448 | 1.469 | 1.421 | 1.418 | 1.402 | 1.398 |
| bike | 0.295 | 0.302 | 0.282 | 0.274 | 0.262 | 0.261 | 0.216 | 0.219 | 0.211 | 0.208 | 0.203 | 0.201 |
| crab | 0.286 | 0.288 | 0.258 | 0.252 | 0.242 | 0.239 | 0.226 | 0.230 | 0.224 | 0.222 | 0.219 | 0.217 |
| insurance | 0.511 | 0.512 | 0.473 | 0.462 | 0.467 | 0.462 | 0.384 | 0.385 | 0.382 | 0.381 | 0.379 | 0.378 |

## C.3 ANALYSIS OF ITERATIVE GAINS IN FEATURE ENGINEERING (ADDITIONAL DETAILS)

Figure C.1a to Figure C.1g compare the performance trajectories of the proposed method with two LLM-based baselines (CAAFE and OCTree) in the feature engineering process, using an iteration budget of 50 and MLP as the tabular learner across seven tasks.

## D PROMPT TEMPLATE

The prompt for GPT-4o to propose feature transformation operations consists of two parts:

- **Introduction Prompt**:

```
You are an expert data scientist with deep expertise in feature
    engineering. You have the ability to:
1) Analyze patterns in previous feature performance to guide new
    feature creation
2) Reason about why certain features succeeded or failed
3) Design complementary features that address gaps in the current
    feature set
4) Consider domain knowledge and statistical relationships in your
    feature design
```

- **Instruction Prompt**:

```
Dataset Context:
- Task type: [CLASSIFICATION_OR_REGRESSION]
- Metric: [ROC_AUC_OR_OTHER]
- Columns (name:type): [COLS_WITH_TYPES]
- Target: <TARGET_NAME>
- Notes (missingness, skew, constraints): <DATA_NOTES>

Recent performance feedback: [PERFORMANCE HISTORY]
Remaining iteration budget: [BUDGET]

**Strategic Reasoning**
Based on the performance feedback above, consider:
1. What patterns do you see in the performance history?
2. What types of relationships might be missing from current
    features?
3. How can you build upon successful features while avoiding
    failed approaches?
4. What domain-specific insights can guide your next feature ideas
    ?

**Task**
Suggest up to K complementary NEW features** as a JSON list. Each
    item should include:

  {
    "name": "snake_case_identifier",
    "explanation": "<detailed reasoning: why this feature helps,
    how it builds on feedback>",
    "reasoning": "<strategic thinking: what patterns from history
    inform this choice>",
    "code": "feature = <python expression using df[...] + helper
    ops>",
    "expected_benefit": "<specific hypothesis about how this will
    improve the model>"
  }

**Important Guidelines:**
- Do not suggest features that need label information.
- Learn from rejected features - avoid similar patterns that
    failed
- Build upon successful features - create complementary variations
- You can try to combine multiple (N > 2) features to create a new
    feature to capture a more complex relationship.
- Ensure features are diverse and capture different aspects of the
    data
- Provide specific, actionable reasoning for each feature choice
- For the reasoning process and expected benefit analysis, be your
    best to be concise and clear.

Return ONLY the JSON list.
```

The prompt for GPT-4o to simulate as the human expert to provide the preference feedback consists of two parts:

- **Introduction Prompt**:

```
You are a senior ML scientist specializing in tabular feature
    engineering and feature evaluation. Given dataset context and
    SHAP-based feature importances from a baseline model, your goal
     is to judge which of two candidate feature operations is more
    likely to improve the downstream metric when added to the
    current pipeline. Avoid label leakage; prefer complementary,
    non-redundant transformations to improve the model performance.
```

- **Instruction Prompt**:

```
Task: Choose the more promising feature operation between A and B

Dataset:
- Task type: [CLASSIFICATION_OR_REGRESSION]
- Metric: [ROC_AUC_OR_OTHER]
- Columns (name:type): [COLS_WITH_TYPES]
- Target: <TARGET_NAME>
- Notes (missingness, skew, constraints): <DATA_NOTES>

Baseline model + SHAP:
- Base model: <MODEL_NAME>
- Top SHAP features (name:score): <[(f1, s1), (f2, s2), ...]>

Recent performance feedback: [PERFORMANCE HISTORY]
Remaining iteration budget: [BUDGET]

Candidates:
A:
- name: <A_NAME>
- code: <A_CODE_SNIPPET_USING_df['...']>
- rationale: <WHY_THIS_MIGHT_HELP>

B:
- name: <B_NAME>
- code: <B_CODE_SNIPPET_USING_df['...']>
- rationale: <WHY_THIS_MIGHT_HELP>

Decision instructions:
- Prefer features that:
  1) Leverage high-SHAP columns sensibly (monotone transforms,
    interactions, ratios/differences, bins);
  2) Complement accepted features (diversity > redundancy);
  3) Are robust to outliers/missingness and unlikely to leak
    labels.
- Penalize features that:
  a) Duplicate existing ones; b) Are overly noisy/fragile;
Output format (JSON only):
{
  "choice": "A" | "B" ,
}
Return ONLY the JSON object.
```

# E EXAMPLES OF LLM-PROPOSED FEATURE OPERATIONS

Below we provide several examples of LLM-proposed feature transformation operations selected by our algorithm. Each block shows the feature name together with its corresponding Python expression.

DAYS_SINCE_FIRST_EVENT_WEIGHTED

```
feature = (0.5 * df['days_since_first_event_xxxxx_event_data']
          + 0.5 * df['days_since_first_event_yyyyy_event_data'])
```

WIFI_CLEANLINESS_BOOKING

```
feature = df['Inflight wifi service'] * df['Cleanliness'] * df['Ease of
    Online booking']
```

DIGITAL_EXPERIENCE_TENSOR

```
gmean = (df['Inflight wifi service'] * df['Ease of Online booking']
        * df['Online boarding']) ** (1/3)
comfort = np.tanh((df['Seat comfort'] + df['Leg room service']) / 2.0)
feature = (gmean * (df['Cleanliness'] ** 0.5)) * comfort
```

BUSINESS_TRAVEL_CLEANLINESS_COMFORT

```
feature = (df['Type of Travel'] == 'Business travel') * \
          df[['Cleanliness', 'Seat comfort', 'Leg room service']].mean(
    axis=1)
```

AGE_WEIGHTED_HEALTH_INTERACTION

```
feature = (df['Age'] * (df['HighBP'] + df['HighChol'] + df['
    HeartDiseaseorAttack'])) \
          / (1 + df['Smoker'] * df['BMI'])
```

LIFESTYLE_RISK_BALANCE_ENHANCED

```
feature = (df['Fruits'] + df['Veggies'] + df['PhysActivity']) / (
    df['Smoker'] + df['HvyAlcoholConsump'] + df['NoDocbcCost'] + 1
)
```

ACTIVITY_DIET_BALANCE

```
feature = (df['Fruits'] + df['Veggies'] + df['PhysActivity']) / (
    df['Smoker'] + df['HvyAlcoholConsump'] + 1
)
```

DIFF_WALK_HEALTH_INTERPLAY

```
feature = df['DiffWalk'] * df['BMI']
```

MULTI_AXIS_RISK_COMPOSITE

```
feature = (
    (df['Age'] * (df['HighBP'] + df['HighChol'] + df['
    HeartDiseaseorAttack']))
    / (1 + df['Smoker'] * (1 + df['BMI'])))
) * (
    (df['Fruits'] + df['Veggies'] + df['PhysActivity'] + 1)
    / (1 + df['HvyAlcoholConsump'] + df['NoDocbcCost'])
)
```

