# OpenReview forum: "Human-LLM Collaborative Feature Engineering for Tabular Data"
_ICLR.cc/2026/Conference — ICLR 2026 Poster_

### Official Review · Reviewer_abCY · 2025-10-20

**Soundness:** 3
**Presentation:** 2
**Contribution:** 3
**Rating:** 2
**Confidence:** 5

**Summary:**

This paper proposes LLM-based feature engineering methods for tabular learning. Specifically, while letting LLM to explore operation candidates, the proposed method evaluates such generated operation rule by human experts, if available. While providing strong empirical evaluations, the authors also provide some theoretical aspects of the proposed method.

**Strengths:**

1. The authors provided strong evaluation results, and also provided some theoretical explanations of their approach.

2. Writing is easy to follow.

3. The evaluation results are based on multiple datasets.

**Weaknesses:**

1. What is the novel point of the proposed method, compared to the existing methods? I acknowledge that involving human experts might have some pros on doing feature engineering, and also agree that it will definitely improve the performance. However, still, involving human experts may cause some scalability issues. In other words, the authors provided the experimental results where the LLM-generated operations are evaluated by LLM itself. In this case, what advantages does the author's method have, compared to only evaluating target AI models (like MLP or XGBoost)?

2. How were the human experts selected? Did the authors act as a human expert? If not, how much did it cost to hire such human experts?

3. It would be better to show the cost analysis compared to baseline methods.

4. The datasets used in the experiments actually have just a few columns. If the table has millions of columns, is the proposed method scalable? Even in this case, is it possible to involve human experts?

5. How can such an approach be expanded when the column names do not exist? For example, for some financial datasets, the column names and values can be anonymized because of some privacy issues.

6. There are some missing citations. For example, P2T [1] is one of the notable work which leverages LLMs for tabular learning.

[1] Nam et al., Tabular Transfer Learning via Prompting LLMs, COLM 2024

**Questions:**

See Weaknesses.

---

> ### Author Response · Authors · 2025-11-20
> **Rebuttal by Authors**
>
> Thank you for your review! We hope the responses below satisfactorily address your concerns and questions, and we are open to discussing any further concerns you may have.
>
> > **novelty of the proposed method**
> > *What is the novel point of the proposed method, compared to the existing methods? I acknowledge that involving human experts might have some pros on doing feature engineering ...... In this case, what advantages does the author's method have, compared to only evaluating target AI models (like MLP or XGBoost)?*
>
> We appreciate the reviewer’s question regarding the novelty of our method. We would first like to clarify a potential misunderstanding in the reviewer’s comment: in our framework, the LLM is never used to directly evaluate the quality of a feature transformation. Instead, the LLM is used solely to propose diverse candidate transformation operations based on its internal heuristics and understanding of the task. The actual selection of which operation to evaluate next is determined by our Bayesian surrogate modeling of the utility and uncertainty of each candidate. The role of the LLM in our system is strictly limited to proposing candidate feature transformations, rather than assessing their usefulness. We apologize if this aspect was not made sufficiently clear in the submitted version.
>
> Building on this clarification, we then would like to clarify how our approach differs from and advances beyond current LLM-based feature engineering baselines. Existing methods such as CAAFE and OCTree tightly couple the feature-operation generation and selection processes. In each iteration of feature engineering, the LLM produces a single “best guess” transformation based on its internal heuristics, and this transformation is then directly evaluated by the downstream model. Such a design heavily relies on LLM heuristics and often leads to local optima. As shown in Figure 1 and Figure C.1 in the manuscript, both CAAFE and OCTree tend to plateau early because the LLM repeatedly proposes low-yield transformations that align with its own prior biases. These methods cannot reason about the relative utility of different candidates or the uncertainty of unexplored ones. Based on this observation, our first contribution is to fully decouple the feature-operation proposal and selection processes. In our design, LLMs are used solely to propose diverse transformation candidates, while the selection is guided by an explicit Bayesian surrogate model that provides both utility and uncertainty estimates for each operation. This architecture fundamentally differs from prior work and already yields substantial performance gains over all baselines even without any human feedback (Table 1 and Table C.2). However, we also identify that at the early stage of feature engineering, when only a few transformations have been evaluated, the Bayesian surrogate may be underfitted and unable to provide reliable feature operation utility estimation. Therefore, we introduce a selective preference-feedback mechanism that queries human feedback only when the surrogate’s uncertainty is high and additional supervision is expected to be valuable. This mechanism helps correct early-stage estimation errors without relying on human input throughout the process, and we show that it further improves performance while requiring only a small number of queries.
>
> > **scalability of the proposed method**
> > *However, involving human experts may cause some scalability issues.*
>
> Regarding the reviewer’s concerns over the scalability of incorporating human experts, we first want to make a note that human feedback is not required for our method to outperform existing approaches and is used only selectively when the surrogate’s uncertainty is high. As Table 1 and Table C.2 show, the version of our method without any human involvement already consistently surpasses all LLM-based and non-LLM baselines. In addition, when human input is used, it is not required at every iteration; instead, humans mainly observe the process, and feedback is queried only when the expected value of additional information outweighs the cognitive cost. In practice, under a budget of 50 iterations for feature engineering, our algorithm requests human preferences only around 7 times on average, which translates into fewer than 15 percent of 50 rounds. Furthermore, in our real-user study on the flight-satisfaction dataset, participants using our framework achieved higher final performance, completed the task more efficiently, and reported substantially lower cognitive load compared to baseline conditions. These results demonstrate that the method scales not only computationally but also in terms of human effort, and that incorporating human experts in a selective manner does not hinder scalability in realistic settings.

---

> ### Author Response · Authors · 2025-11-20
> **Rebuttal by Authors (Continued)**
>
> > *The datasets used in the experiments actually have just a few columns. If the table has millions of columns, is the proposed method scalable? Even in this case, is it possible to involve human experts?*
>
> We appreciate the reviewer’s question regarding scalability with respect to the number of initial columns. To further illustrate the computational scalability of our framework, we measured the runtime of each component in a single feature-engineering iteration. Each round of our method consists of four steps: (1) calling the LLM to generate candidate transformations, (2) fitting the BNN surrogate model, (3) computing UCB scores, and (4) evaluating the selected transformation using the downstream tabular model. Among these, steps 2–3 correspond to the core contributions of our method, while steps 1 and 4 are shared by all LLM-based pipelines. To avoid confounding effects from dataset semantics, we constructed synthetic binary classification datasets and varied (a) the number of initial feature columns and (b) the number of data instances to directly measure how each component of the pipeline scales with feature dimensionality and dataset size. All feature columns were generated from standard Gaussian distributions, labels from a Bernoulli distribution, the MLP model as the downstream evaluator, and and all LLM calls used GPT-4o API.
>
> We first fixed the dataset size to 10,000 samples and varied the number of initial feature columns from 10 to 10,000. As shown in Table A, the surrogate model fitting time and UCB score computation time increase only mildly as the number of features grows, while the downstream model evaluation grows more noticeably. Even with as many as 10,000 initial features, the surrogate and UCB components together remain about 2.2% of the total runtime.
>
> **Table A. Runtime breakdown when varying the number of initial features (n_samples = 10,000).**
>
> | #Features | LLM Mean (s) | Surrogate Mean (s) | UCB Mean (s) | Eval Mean (s) |
> |-----------|--------------|---------------------|--------------|----------------|
> | 10        | 1.820303     | 0.166093            | 0.006227     | 1.796845       |
> | 50        | 1.820303     | 0.162171            | 0.004734     | 1.247297       |
> | 100       | 1.820303     | 0.190383            | 0.004638     | 1.781571       |
> | 1,000     | 1.820303     | 0.201621            | 0.008805     | 8.040899       |
> | 10,000    | 1.820303     | 0.570452            | 0.018869     | 23.431735      |
>
> Next, we fixed the number of initial features to 100 and varied the dataset size from 1,000 to 100,000 instances. As shown in Table B, the surrogate fitting and UCB computation times remain nearly constant across all sample sizes and contribute only a small percentage of the total runtime, because both operate at the feature-operation level and are independent of the number of rows. In contrast, the downstream evaluation time increases with dataset size. For example, even at 100,000 samples, the surrogate and UCB components together account for only about 1.3 percent of the total computation.
>
> **Table B. Runtime breakdown when varying the number of samples (n_features = 100).**
>
> | #Samples | LLM Mean (s) | Surrogate Mean (s) | UCB Mean (s) | Eval Mean (s) |
> |----------|--------------|---------------------|--------------|----------------|
> | 1,000    | 1.820303     | 0.173182            | 0.005338     | 0.289455       |
> | 5,000    | 1.820303     | 0.179113            | 0.005404     | 0.887437       |
> | 10,000   |1.820303     | 0.232185            | 0.006151     | 1.465134       |
> | 50,000   | 1.820303     | 0.182320            | 0.005778     | 5.222195       |
> | 100,000  | 1.820303     | 0.179800            | 0.005241     | 10.651274      |
>
> As shown in our analysis, the cost of our method grows only mildly with the number of initial columns: even when increasing the number of features by a factor of 1,000 (from 10 to 10,000), the surrogate fitting and UCB computation still remain under 3% of the total runtime. This demonstrates that the additional overhead introduced by our framework stays small even as the feature dimensionality explods. Regarding the hypothetical case where a table contains millions of columns,  we want to make a note that our proposed method operates in the feature transformation space rather than the raw column space. When providing preference feedback, human experts are never required to examine all raw features. Human feedback is queried only to compare two proposed transformations at a time, and each comparison typically involves only a few relevant columns rather than the full feature space. The number of such queries also remains very small in practice, approximately 7 per 50 iterations on average.
>
> Taken together, these observations show that our framework scales well under realistic column count scenarios and that the cost of involving human experts does not grow with the total number of raw features.

---

> ### Author Response · Authors · 2025-11-20
> **Rebuttal by Authors (Continued)**
>
> > *How were the human experts selected? Did the authors act as a human expert? If not, how much did it cost to hire such human experts?*
>
> All participants were recruited from ML engineers/scientists and AI/ML graduate students in our organization voluntarily, and the authors did not serve as the human experts. They were compensated in the form of a lunch that we provided for them, so the overall cost of involving human experts was minimal (we simply bought them lunch :)).
>
> > *How can such an approach be expanded when the column names do not exist? For example, for some financial datasets, the column names and values can be anonymized because of some privacy issues.*
>
> Regarding the reviewer’s concern about whether our approach can work when column names are anonymized or unavailable, we clarify that the proposed method does not require column-name semantics. When our appoach models the utility and uncertainty of different feature operations, we use only the column ID and the embedding of the feature operation as the input. The column names are not used.
> In our main experiments, we followed the prompting setup used in prior work such as CAAFE [1] and OCTree [2], where the LLM responsible for generating feature operations is provided with column names when they are available, together with other dataset metadata such as the prediction objective, feature types, and representative values. However, column names are not required by our framework. When column names are anonymized or unavailable, the LLM can simply operate using column indices and structured metadata to propose transformation candidates.
> To further validate that our method does not rely on column-name semantics and continues to outperform existing approaches even when names are removed, we conducted additional experiments on additional four datasets after fully anonymizing all column names by replacing them with generic identifiers such as “col_1”, “col_2”, and so on. In these experiments, we used GPT-4o as the backbone model for generating candidate transformations and an MLP classifier as the downstream tabular model. The feature-engineering iteration budget was set to 50 for all methods. For the Ours (w/ human) setting, GPT-4o was also used to simulate the human expert and provide preference feedback.
> The results are shown in Table A. Across all four tasks, anonymizing column names does not degrade performance, and our method consistently outperforms both CAAFE and OCTree. These findings confirm that our framework does not depend on feature-name semantics and remains effective and robust in privacy-sensitive environments where column names are removed or obfuscated.
>
> **Table A. Performance on anonymized-column datasets.**
>
> | Method              | Loan  | Stroke | Student Performance | Fraud |
> |---------------------|-------|--------|----------------------|-------|
> | CAAFE               | 0.939 | 0.927  | 0.784                | 0.879 |
> | OCTree              | 0.945 | 0.938  | 0.801                | 0.906 |
> | Ours (w/o human)    | 0.952 | 0.946  | 0.807                | 0.914 |
> | Ours (w/ human)     | 0.957 | 0.949  | 0.812                | 0.918 |
>
> > *There are some missing citations. For example, P2T [1] is one of the notable work which leverages LLMs for tabular learning.*
>
> We thank the reviewer for pointing out the missing citations. We have followed your suggestion and included P2T [3] and the paper STUNT [4] in our manuscript. If the reviewer has additional recommendations for relevant citations that we may have overlooked, we would be grateful if you could list them, and we will incorporate them into the camera-ready version as well.
>
> [1] Hollmann et al., Large Language Models for Automated Data Science: Introducing CAAFE for Context-Aware Automated Feature Engineering, NeurIPS 2023
>
> [2] Nam et al., Optimized Feature Generation for Tabular Data via LLMs with Decision Tree Reasoning, NeurIPS 2024
>
> [3] Nam et al., Tabular Transfer Learning via Prompting LLMs, COLM 2024
>
> [4]  Nam et al., STUNT: Fewshot tabular learning with self-generated tasks from unlabeled tables, ICLR 23

---

> ### Comment · Reviewer_abCY · 2025-11-24
>
> Thanks for the rebuttal including clarification on your method. I acknowledge that I slightly had some misunderstandings, therefore read the paper and the other reviewer's comments comprehensively. I think this paper can contribute to ICLR community, therefore having re-evaluated the score.
>
> In addition, some analysis during the rebuttal (e.g., scale-up experiments) was really interesting. I highly encourage the authors to include such results in the final manuscript.

---

> > ### Author Response · Authors · 2025-11-24
> >
> > Thank you for taking the time to review our paper again. We’ll follow your suggestions and add the additional analyses in the final manuscript. Have a great day! :)

---

### Official Review · Reviewer_oY9U · 2025-10-28

**Soundness:** 2
**Presentation:** 3
**Contribution:** 3
**Rating:** 6
**Confidence:** 3

**Summary:**

The paper proposes a human-LLM collaborative framework for tabular learning in which the generation of candidate features and selection are decoupled for better automatic feature engineering. Specifically, LLMs are used solely for the generation of operation candidates, while a separate surrogate model guided by Bayesian optimization is used to model the utility and uncertainty of those candidates for selection. As accurate estimation of the utility can often be difficult, the framework selectively elicits input from human experts. Empirical results using GPT-4o as a simulated human demonstrate promising performance compared to prior work.

**Strengths:**

1. Clearly motivated framework. The authors rightly point out the conceptual limitation of existing LLM-based feature engineering approaches, which is that an LLM is used as a black-box optimizer for both proposing and selecting new features. Instead, the paper proposes using LLMs solely for generation with the selection guided by a Bayesian neural network as a surrogate model for utility and uncertainty estimation.

2. Principled human-in-the-loop mechanism. For more accurate estimate utility, the paper adds a mechanism to elicit human expert preference feedback. In order to make more effective use of human expertise, the propose mechanism selectively request human feedback based on the estimated potential for improved selection ("overlap" and "uncertainty").

**Weaknesses:**

1. Missing ablations on the backbone model. The authors primarily evaluate GPT-4o as the backbone model for LLM-based feature engineering methods. It would be useful to analyze how much the joint feature proposal and selection is an issue depending on the backbone model, e.g., comparing open vs. closed models, different model sizes, and reasoning-enabled vs. non-reasoning models.

2. Missing experiments with humans of varying levels of expertise. In the main experiments, GPT-4o is used as a simulated human -- which is likely comparable to a human expert -- in which case the method sees additional improvements compared to the version without it. It would be useful to better understand what level of human expertise (e.g., using noisy preference) is required to achieve these additional gains (or, potentially loss in performance with poor feedback).

3. Evaluation of scalability. It is unclear how much computational cost is involved in fitting BNN surrogate, calculating UCB values for every candidate, etc. Additional results on scalability with datasets of varying sample sizes and numbers of initial features would provide greater insight.

**Questions:**

Q. Have the authors experimented with alternative embeddings for the surrogate model before deciding on $[\phi_\text{embedding}, \phi_\text{column}]$?

Q. How does the method scale to datasets with varying numbers of initial samples and features?

---

> ### Author Response · Authors · 2025-11-20
> **Rebuttal by Authors**
>
> Thank you for your review! We hope the responses below satisfactorily address your concerns and questions, and we are open to discussing any further concerns you may have.
>
>
> > Missing ablations on the backbone model...
>
> We thank the reviewer for raising this point. To examine how the choice of backbone model affects the performance of our framework, we conducted an additional experiment using three LLMs beyond GPT-4o: DeepSeek-v3 (open-source), GPT-3.5-turbo (smaller, non-reasoning model), and GPT-5 (larger, reasoning-capable closed model). These models span different families and capability levels, allowing us to probe the stability of our approach across heterogeneous generators. Tables A and B report the average AUROC on the 13 datasets used in Table 1 of the main manuscript.
> Across all three newly added LLMs, the performance trends remain consistent with those observed in the main paper using GPT-4o. Even when the generator is relatively weak (e.g., GPT-3.5-turbo), our method effectively filters out poor or redundant feature transformations, and the overall performance stays above all baseline feature-engineering systems. Stronger generators such as GPT-5 and DeepSeek-v3 naturally yield higher-quality proposals and thus higher absolute accuracy, but the relative improvement of our method over the baselines is stable across every backbone. When a human collaborator is available, incorporating preference feedback improves performance for all generators.
> Overall, these results indicate that our feature-engineering framework is robust to the choice of backbone LLM and does not depend on any specific model family or scale.
> We have incorporated this new backbone ablation analysis into the revised manuscript.
>
> **Table A. Average AUROC on 13 datasets with different LLMs and MLP as the downstream tabular model.**
>
> | Model        | OpenFE  | AutoGluon | CAAFE  | OCTree | Ours (w/o human) | Ours (w/ human) |
> |--------------|---------|-----------|--------|--------|-------------------|------------------|
> | Deepseek-v3  | 0.8348  | 0.8345    | 0.8490 | 0.8550 | 0.8610            | 0.8640           |
> | GPT3.5-turbo | 0.8348  | 0.8345    | 0.8315 | 0.8420 | 0.8460            | 0.8510           |
> | GPT4o        | 0.8348  | 0.8345    | 0.8430 | 0.8470 | 0.8530            | 0.8550           |
> | GPT5         | 0.8348  | 0.8345    | 0.8554 | 0.8575 | 0.8590            | 0.8651           |
>
>
> **Table B. Average AUROC on 13 datasets with different LLMs and XGBoost as the downstream tabular model.**
>
> | Model        | OpenFE  | AutoGluon | CAAFE  | OCTree | Ours (w/o human) | Ours (w/ human) |
> |--------------|---------|-----------|--------|--------|-------------------|------------------|
> | Deepseek-v3  | 0.8558  | 0.8535    | 0.8660 | 0.8731 | 0.8820            | 0.8860           |
> | GPT3.5-turbo | 0.8558  | 0.8535    | 0.8521 | 0.8603 | 0.8654            | 0.8710           |
> | GPT4o        | 0.8558  | 0.8535    | 0.8590 | 0.8670 | 0.8740            | 0.8740           |
> | GPT5         | 0.8558  | 0.8535    | 0.8712 | 0.8774 | 0.8801            | 0.8870           |
>
> > Missing experiments with humans of varying levels of expertise.
>
> To understand how humans with different levels of expertise would affect the performance of the proposed method, we conducted an additional robustness experiment where we explicitly injected noise into GPT-4o’s pairwise preferences (random flip probability 0.1–0.5). Specifically, for each queried pair, we randomly flipped GPT-4o’s original preference with probability $\epsilon \in \\{0.1,0.2,0.3,0.4,0.5\\}$.. We used GPT-4o as the backbone LLM for generating feature operations, fixed the iteration budget to 50, and evaluated all variants using an MLP classifier. The results in Table C show that our method remains highly stable under the preference feedback with different nosiy level. Across most of datasets, our framework continues to outperform both OCTree and the “w/o human” version even with the nosiy feedback..
>
> **Table C. Performance of our method under noisy preference feedback.**
>
> | Dataset   | OCTree | Ours w/o Human | $\epsilon$=0.0 | $\epsilon$=0.1 | $\epsilon$=0.2 | $\epsilon$=0.3 | $\epsilon$=0.4 | $\epsilon$=0.5 |
> |-----------|--------|----------------|-------|-------|-------|-------|-------|-------|
> | Flight    | 94.8   | 96.9           | 97.3  | 97.3  | 97.2  | 97.1  | 97.0  | 97.0  |
> | Titanic   | 86.5   | 86.8           | 87.0  | 87.0  | 86.9  | 86.9  | 86.9  | 86.7 |
> | Wine      | 78.2   | 78.5           | 78.7  | 78.7  | 78.6  | 78.6  | 78.5  | 78.4  |
> | Diabetes  | 82.8   | 83.0           | 83.0  | 83.0  | 83.0  | 82.9  | 83.0  | 82.9  |
> | Adult     | 90.9   | 91.3           | 91.4  | 91.4  | 91.3  | 91.2  | 91.3  | 91.2  |
> | Heart     | 93.1   | 93.4           | 93.6  | 93.6  | 93.5  | 93.4  | 93.4  | 93.4  |
> | Conversion  | 91.1   | 92.6           | 92.9  | 92.9  | 92.9  | 92.8  | 92.8  | 92.8  |

---

> ### Author Response · Authors · 2025-11-20
> **Rebuttal by Authors (Continued)**
>
> > **Q. Scalability concerns.**
> >
> > (1) *How does the method scale to datasets with varying numbers of initial samples and features?*
> >
> > (2) *Evaluation of scalability. It is unclear how much computational cost is involved in fitting the BNN surrogate, calculating UCB values for every candidate, etc. Additional results on scalability with datasets of varying sample sizes and numbers of initial features would provide greater insight.*
>
>
> To analyze the computational cost for each component of  the proposed method, we measured the runtime of each component in a single feature-engineering iteration. Each round of our method consists of four steps:
>
> 1. calling the LLM to generate candidate feature transformations,
> 2. fitting the BNN surrogate model using the accumulated observations,
> 3. computing the UCB score for each proposed candidate, and
> 4. evaluating the selected transformation using the downstream tabular model.
>
>
>  Among these components, steps 2–3 correspond to the core contribution of our method, while steps 1 and 4 are shared by all LLM-based feature engineering pipelines.
> To avoid confounding effects from dataset semantics, we constructed synthetic binary classification datasets and systematically varied (a) the number of initial feature columns and (b) the number of data instances to directly measure how each component of the pipeline scales with feature dimensionality and dataset size. All feature columns were generated from standard Gaussian distributions, labels from a Bernoulli distribution, an MLP model was used as the downstream evaluator, and all LLM calls were made using the GPT-4o API.
> We first fixed the dataset size to 10,000 samples and varied the number of initial feature columns from 10 to 10,000. As shown in Table A, the surrogate model fitting time and UCB score computation time increase only mildly as the number of features grows, while the downstream model evaluation grows more noticeably. The LLM call time remains constant, since it depends only on the prompt size rather than the data size. Even with as many as 10,000 initial features, the surrogate and UCB components together remain about 2.2% of the total runtime.
>
> **Table A. Runtime breakdown when varying the number of initial features (n_samples = 10,000).**
>
> | #Features | LLM Mean (s) | Surrogate Mean (s) | UCB Mean (s) | Eval Mean (s) |
> |-----------|--------------|---------------------|--------------|----------------|
> | 10        | 1.820303     | 0.166093            | 0.006227     | 1.796845       |
> | 50        | 1.820303     | 0.162171            | 0.004734     | 1.247297       |
> | 100       | 1.820303     | 0.190383            | 0.004638     | 1.781571       |
> | 1,000     | 1.820303     | 0.201621            | 0.008805     | 8.040899       |
> | 10,000    | 1.820303     | 0.570452            | 0.018869     | 23.431735      |
>
> Next, we fixed the number of initial features to 100 and varied the dataset size from 1,000 to 100,000 instances. As shown in Table B, the surrogate fitting and UCB computation times remain nearly constant across all sample sizes and contribute only a small percentage of the total runtime, because both operate at the feature-operation level and are independent of the number of rows. In contrast, the downstream evaluation time increases with dataset size, as it requires training the MLP on the full dataset. For example, even at 100,000 samples, the surrogate and UCB components together account for only about 1.3 percent of the total computation.
>
> The LLM call time again remains constant.
>
> **Table B. Runtime breakdown when varying the number of samples (n_features = 100).**
>
> | #Samples | LLM Mean (s) | Surrogate Mean (s) | UCB Mean (s) | Eval Mean (s) |
> |----------|--------------|---------------------|--------------|----------------|
> | 1,000    | 1.820303     | 0.173182            | 0.005338     | 0.289455       |
> | 5,000    | 1.820303     | 0.179113            | 0.005404     | 0.887437       |
> | 10,000   |1.820303     | 0.232185            | 0.006151     | 1.465134       |
> | 50,000   | 1.820303     | 0.182320            | 0.005778     | 5.222195       |
> | 100,000  | 1.820303     | 0.179800            | 0.005241     | 10.651274      |
>
> Taken together, these results show that our method scales effectively in both directions.
> When fixing the number of features and increasing sample size from 1,000 to 100,000, the surrogate model  plus UCB components consistently contribute only about 1–2 percent of total runtime.
> When fixing sample size and increasing the number of features from 10 to 10,000, they remain below 3 percent, while the downstream model evaluation accounts for more than 95 percent of total cost.
> Overall, the core components introduced by our method add minimal extra computation cost and scale smoothly to large datasets and high-dimensional feature spaces. We also included the scalability analysis of this part in the revised manuscript.

---

> > ### Author Response · Authors · 2025-11-20
> > **Rebuttal by Authors (Continued)**
> >
> > > Have the authors experimented with alternative embeddings for the surrogate model before deciding on ....
> >
> > Thanks for the question. Before settling on the final embedding design for the surrogate model, we also experimented with several alternative embeddings.
> > First, we tested a feature-type analysis embedding that encodes whether the generated feature is numeric or categorical, its sparsity, cardinality, variance, and skewness. However, we found that adding this component often caused the surrogate model to overfit. In practice, the utility and uncertainty estimates collapsed to nearly identical values across different operations, making the acquisition function ineffective. Removing this component resolved the issue.
> > We also experimented with a hand-crafted statistical feature vector that describes the complexity of the LLM-generated pandas code to generate the new feature column, including the number of source columns used, the number of transformation steps, and operation-type indicators (log, sqrt, groupby, cut, addition, subtraction, multiplication, division, median, max, min). In our studies, these signals did not provide complementary information once text embeddings were included. The semantic embeddings already captured these patterns, making the handcrafted vector redundant while adding unnecessary noise.
> > Based on these observations, we selected the final embedding design reported in the paper.

---

> ### Author Response · Authors · 2025-11-20
> **Rebuttal by Authors (Continued)**
>
> > How sensitive is the framework to different generators (GPT-4o vs GPT-3.5 vs open-source LLMs)? Will low-quality LLM-generated candidates drag down the entire selector?The paper ...
>
> We thank the reviewer for raising this point. To examine how the choice of backbone model affects the performance of our framework, we conducted an additional experiment using three LLMs beyond GPT-4o: DeepSeek-v3 (open-source), GPT-3.5-turbo (smaller, non-reasoning model), and GPT-5 (larger, reasoning-capable closed model). These models span different families and capability levels, allowing us to probe the stability of our approach across heterogeneous generators. Tables A and B report the average AUROC on the 13 datasets used in Table 1 of the main manuscript.
> Across all three newly added LLMs, the performance trends remain consistent with those observed in the main paper using GPT-4o. Even when the generator is relatively weak (e.g., GPT-3.5-turbo), our method effectively filters out poor or redundant feature transformations, and the overall performance stays above all baseline feature-engineering systems. Stronger generators such as GPT-5 and DeepSeek-v3 naturally yield higher-quality proposals and thus higher absolute accuracy, but the relative improvement of our method over the baselines is stable across every backbone. When a human collaborator is available, incorporating preference feedback improves performance for all generators. Overall, these results indicate that our feature-engineering framework is robust to the choice of backbone LLM and does not depend on any specific model family or scale.
> We further examined the intrinsic quality of the feature-transformation operations generated by different LLMs. We first considered potential label leakage. Since the prompt explicitly instructs the model not to reference the target column or any transformation of it, all LLMs adhered to this constraint consistently, and the leakage rate was effectively zero for every generator. We then evaluated the similarity of the proposed transformations across LLMs. Using the same embedding representation fed into our BNN surrogate, we computed the cosine similarity between each generated feature and the existing feature set. GPT-3.5-turbo showed slightly higher similarity scores on average, although the differences across models were modest. To better interpret these results, we conducted a qualitative analysis of the generated operations. GPT-3.5-turbo tended to repeatedly produce simple, shallow transformations—typically involving only one or two columns—while rarely exploring more compositional or multi-step feature interactions. In contrast, stronger generators such as GPT-5 and DeepSeek-v3 more often synthesized deeper, multi-column transformations and demonstrated a clearer ability to build on the operations proposed in earlier rounds.
> We have incorporated this new backbone ablation analysis into the revised manuscript.
>
> **Table A. Average AUROC on 13 datasets with different LLMs and MLP as the downstream tabular model.**
> | Model        | OpenFE  | AutoGluon | CAAFE  | OCTree | Ours (w/o human) | Ours (w/ human) |
> |--------------|---------|-----------|--------|--------|-------------------|------------------|
> | Deepseek-v3  | 0.8348  | 0.8345    | 0.8490 | 0.8550 | 0.8610            | 0.8640           |
> | GPT3.5-turbo | 0.8348  | 0.8345    | 0.8315 | 0.8420 | 0.8460            | 0.8510           |
> | GPT4o        | 0.8348  | 0.8345    | 0.8430 | 0.8470 | 0.8530            | 0.8550           |
> | GPT5         | 0.8348  | 0.8345    | 0.8554 | 0.8575 | 0.8590            | 0.8651           |
>
>
> **Table B. Average AUROC on 13 datasets with different LLMs and XGBoost as the downstream tabular model.**
>
> | Model        | OpenFE  | AutoGluon | CAAFE  | OCTree | Ours (w/o human) | Ours (w/ human) |
> |--------------|---------|-----------|--------|--------|-------------------|------------------|
> | Deepseek-v3  | 0.8558  | 0.8535    | 0.8660 | 0.8731 | 0.8820            | 0.8860           |
> | GPT3.5-turbo | 0.8558  | 0.8535    | 0.8521 | 0.8603 | 0.8654            | 0.8710           |
> | GPT4o        | 0.8558  | 0.8535    | 0.8590 | 0.8670 | 0.8740            | 0.8740           |
> | GPT5         | 0.8558  | 0.8535    | 0.8712 | 0.8774 | 0.8801            | 0.8870

---

### Official Review · Reviewer_M3JD · 2025-10-31

**Soundness:** 2
**Presentation:** 3
**Contribution:** 3
**Rating:** 4
**Confidence:** 2

**Summary:**

This paper addresses the limitations of existing feature engineering methods for tabular learning and proposes a Human-LLM collaborative feature engineering framework.The framework decouples the "operation proposal" and "operation selection" processes: LLMs only generate diverse feature transformation candidates based on task understanding, while selection is guided by explicit modeling of the utility and uncertainty of each candidate. For scenarios where utility estimation is inaccurate (especially in early iterations), the framework designs a selective human feedback mechanism—it elicits human experts’ pairwise preference feedback (comparing which operations are more promising) only when the potential gain of feedback outweighs the cognitive cost, thereby improving selection accuracy .In terms of technical implementation, a Bayesian Neural Network (BNN) is used as a surrogate model to approximate the black-box utility function of features, and the Upper Confidence Bound (UCB) strategy is adopted to balance exploitation and exploration when human feedback is unavailable.

**Strengths:**

（1）Innovative Method Design with Clear Targets:The core idea of decoupling LLM’s "operation proposal" and "operation selection" effectively addresses the key limitation of existing LLM-powered methods (i.e., LLMs acting as black-box optimizers). By introducing explicit utility and uncertainty modeling, the framework avoids blind exploration of low-yield operations, and the selective human feedback mechanism balances the value of human expertise and cognitive cost.
（2）Comprehensive Experimental Validation and Multi-Dimensional Evaluation:Diverse Dataset Coverage: The experiments include public datasets (Kaggle, UCI) and a proprietary enterprise dataset, avoiding overfitting to public data biases and enhancing the generalization of results.In addition to downstream model performance (AUROC for classification, normalized RMSE for regression), the study incorporates a user study using the NASA-TLX scale to measure cognitive load, which fully reflects the framework’s practical value in human-AI collaboration scenarios.
（3）Technical Implementation and Transparency:The paper provides clear mathematical formulations for core components (e.g., utility function, surrogate model training objective, UCB calculation), and details key parameter settings. Appendices also include dataset descriptions, prompt templates, and proof of technical lemmas, which is conducive to reproducibility.

**Weaknesses:**

（1）Limitation in Human Feedback Simulation: The "w/ Human" setting in the experiment uses GPT-4o to simulate human experts, rather than recruiting real domain experts for feedback. This may deviate from the actual scenario where human experts rely on domain experience to make judgments, and the authenticity of feedback needs to be further verified.
（2）Insufficient Discussion on Scalability: The paper does not discuss the framework’s performance in ultra-large-scale tabular data scenarios. The BNN surrogate model may face efficiency challenges in high-dimensional feature spaces, and the cost of LLM generating operation candidates may increase significantly.
（3）Lack of Analysis on LLM Candidate Quality: The framework assumes that LLMs can generate high-quality operation candidates, but does not analyze the impact of LLM performance differences (e.g., GPT-4o vs. GPT-3.5, open-source LLMs like LLaMA) on the framework’s final effect. It also does not discuss how to handle low-quality candidates generated by LLMs.

**Questions:**

（1）This paper uses GPT-4o to replace expert scoring. Has consideration been given to how the preferences simulated by GPT-4o differ from those of real domain experts in terms of consistency, noise distribution, and preference biases?
Can the performance gains observed under simulated feedback be replicated with real human feedback? How do feedback latency and fatigue affect the trigger strategy (C1/C2)? Is the cognitive cost threshold γ=4 based on real measurements? How does this threshold vary across different expert groups?
（2）Why choose BNN over (sparse) GP, lightweight ensemble, or incremental update strategies? Is uncertainty estimation under high-dimensional ϕ(e) reliable for BNN?
（3）How sensitive is the framework to different generators (GPT-4o vs GPT-3.5 vs open-source LLMs)? Will low-quality LLM-generated candidates drag down the entire selector?The paper could consider candidate quality metrics such as duplication rate, similarity to existing features, and potential label leakage, and analyze the correlation between these metrics and final performance.

---

> ### Author Response · Authors · 2025-11-20
> **Rebuttal by Authors**
>
> Thank you for your review! We hope the responses below satisfactorily address your concerns and questions, and we are open to discussing any further concerns you may have.
>
>
> > **Human preference feedback**
> > *This paper uses GPT-4o to replace expert scoring. Has consideration been given to how the preferences simulated by GPT-4o differ from those of real domain experts in terms of consistency, noise distribution, and preference biases?
> > Can the performance gains observed under simulated feedback be replicated with real human feedback?*
>
>  We agree with the reviewer that GPT-4o cannot fully replicate the knowledge or judgment of real domain experts. In our synthetic experiments, we intentionally provide GPT-4o with SHAP explanations so that it can ground its preferences in feature-importance signals rather than free-form heuristics, but this still represents only a proxy for human domain expertise. In realistic feature-engineering settings where domain structure is more complex, an LLM-based judge may lack the domain-specific insights that actual experts possess, and thus inevitably introduce additional noise compared with real human feedback. To understand the effect of such imperfect or noisy preference on the performance of the proposed method, we conducted an additional robustness experiment where we explicitly inject noise into GPT-4o’s pairwise preferences (random flip probability 0.1–0.5). Specifically, for each queried pair, we randomly flipped GPT-4o’s original preference with probability $\epsilon \in \\{0.1,0.2,0.3,0.4,0.5\\}$.. We used GPT-4o as the backbone LLM for generating feature operations, fixed the iteration budget to 50, and evaluated all variants using an MLP classifier. The results in Table A show that our method remains highly stable under the preference feedback with different nosiy level. Across most of datasets, our framework continues to outperform both OCTree and the “w/o human” version even with the nosiy feedback. This findings also aligns with our real user study that when the real ML engineers/scientists collaborated with our framework on the flight–satisfaction prediction task, their average final AUROC (~0.976) exceeded LLM-only baselines such as CAAFE and OCTree under the same iteration budget. This provides empirical evidence that actual human expertise yields at least comparable, and often stronge benefits than the simulated expert to provide preference feedback.
>
> **Table A. Performance of our method under noisy preference feedback.**
>
> | Dataset   | OCTree | Ours w/o Human | $\epsilon$=0.0 | $\epsilon$=0.1 | $\epsilon$=0.2 | $\epsilon$=0.3 | $\epsilon$=0.4 | $\epsilon$=0.5 |
> |-----------|--------|----------------|-------|-------|-------|-------|-------|-------|
> | Flight    | 94.8   | 96.9           | 97.3  | 97.3  | 97.2  | 97.1  | 97.0  | 97.0  |
> | Titanic   | 86.5   | 86.8           | 87.0  | 87.0  | 86.9  | 86.9  | 86.9  | 86.7 |
> | Wine      | 78.2   | 78.5           | 78.7  | 78.7  | 78.6  | 78.6  | 78.5  | 78.4  |
> | Diabetes  | 82.8   | 83.0           | 83.0  | 83.0  | 83.0  | 82.9  | 83.0  | 82.9  |
> | Adult     | 90.9   | 91.3           | 91.4  | 91.4  | 91.3  | 91.2  | 91.3  | 91.2  |
> | Heart     | 93.1   | 93.4           | 93.6  | 93.6  | 93.5  | 93.4  | 93.4  | 93.4  |
> | Conversion  | 91.1   | 92.6           | 92.9  | 92.9  | 92.9  | 92.8  | 92.8  | 92.8  |
>
> > **feedback cost & cognitive threshold**
> > *How do latency and fatigue affect C1/C2… how does $\gamma$ vary across expert groups?*
>
> Our current framework does not explicitly model latency or fatigue as separate variables. Instead, these factors are implicitly governed by the cognitive cost parameter \gamma in Condition (C2). A larger $\gamma$ makes the algorithm more conservative and therefore reduces the number of human queries to effectively limit both latency and potential fatigue. Conversely, a smaller \gamma allows more frequent querying when human input is inexpensive. In this work, we set $\gamma = 4$ based on a small pilot with ML practitioners to balance minimal human involvement with minimal performance degradation. And the appropriate values of $\gamma$ may vary across expert groups. For example, domain experts with higher opportunity cost or slower response time may prefer a larger $\gamma$, while high-bandwidth or in-house analysts could operate with a smaller $\gamma$; novice users who provide noisier feedback may also benefit from a higher $\gamma$ to prevent excessive querying. Our framework is agnostic to the specific value, and $\gamma$ can be tuned to reflect the latency tolerance, fatigue sensitivity, and interaction bandwidth of the target user population to maximize the overall feature engineering performance and the user's subjective perceptions.

---

> ### Author Response · Authors · 2025-11-20
> **Rebuttal by Authors (Continued)**
>
> > *Insufficient discussion on scalability: The paper does not discuss the framework’s performance in ultra-large-scale tabular data scenarios ...*
>
> Thanks for raising concerns over the scalability of the proposed method. To further illustrate the computational scalability of our framework, we measured the runtime of each component in a single feature-engineering iteration. Each round of our method consists of four steps:
>
> 1. calling the LLM to generate candidate feature transformations,
> 2. fitting the BNN surrogate model using the accumulated observations,
> 3. computing the UCB score for each proposed candidate, and
> 4. evaluating the selected transformation using the downstream tabular model.
>
>
>  Among these components, steps 2–3 correspond to the core contribution of our method, while steps 1 and 4 are shared by all LLM-based feature engineering pipelines.
> To avoid confounding effects from dataset semantics, we constructed synthetic binary classification datasets and systematically varied (a) the number of initial feature columns and (b) the number of data instances to directly measure how each component of the pipeline scales with feature dimensionality and dataset size. All feature columns were generated from standard Gaussian distributions, labels from a Bernoulli distribution, an MLP model was used as the downstream evaluator, and all LLM calls were made using the GPT-4o API.
> We first fixed the dataset size to 10,000 samples and varied the number of initial feature columns from 10 to 10,000. As shown in Table A, the surrogate model fitting time and UCB score computation time increase only mildly as the number of features grows, while the downstream model evaluation grows more noticeably. The LLM call time remains constant, since it depends only on the prompt size rather than the data size. Even with as many as 10,000 initial features, the surrogate and UCB components together remain about 2.2% of the total runtime.
>
> **Table A. Runtime breakdown when varying the number of initial features (n_samples = 10,000).**
>
> | #Features | LLM Mean (s) | Surrogate Mean (s) | UCB Mean (s) | Eval Mean (s) |
> |-----------|--------------|---------------------|--------------|----------------|
> | 10        | 1.820303     | 0.166093            | 0.006227     | 1.796845       |
> | 50        | 1.820303     | 0.162171            | 0.004734     | 1.247297       |
> | 100       | 1.820303     | 0.190383            | 0.004638     | 1.781571       |
> | 1,000     | 1.820303     | 0.201621            | 0.008805     | 8.040899       |
> | 10,000    | 1.820303     | 0.570452            | 0.018869     | 23.431735      |
>
> Next, we fixed the number of initial features to 100 and varied the dataset size from 1,000 to 100,000 instances. As shown in Table B, the surrogate fitting and UCB computation times remain nearly constant across all sample sizes and contribute only a small percentage of the total runtime, because both operate at the feature-operation level and are independent of the number of rows. In contrast, the downstream evaluation time increases with dataset size, as it requires training the MLP on the full dataset. For example, even at 100,000 samples, the surrogate and UCB components together account for only about 1.3 percent of the total computation.
>
> The LLM call time again remains constant.
>
> **Table B. Runtime breakdown when varying the number of samples (n_features = 100).**
>
> | #Samples | LLM Mean (s) | Surrogate Mean (s) | UCB Mean (s) | Eval Mean (s) |
> |----------|--------------|---------------------|--------------|----------------|
> | 1,000    | 1.820303     | 0.173182            | 0.005338     | 0.289455       |
> | 5,000    | 1.820303     | 0.179113            | 0.005404     | 0.887437       |
> | 10,000   |1.820303     | 0.232185            | 0.006151     | 1.465134       |
> | 50,000   | 1.820303     | 0.182320            | 0.005778     | 5.222195       |
> | 100,000  | 1.820303     | 0.179800            | 0.005241     | 10.651274      |
>
> Taken together, these results show that our method scales effectively in both directions.
> When fixing the number of features and increasing sample size from 1,000 to 100,000, the surrogate model  plus UCB components consistently contribute only about 1–2 percent of total runtime.
> Overall, the core components introduced by our method add minimal extra computation cost and scale smoothly to large datasets and high-dimensional feature spaces. We also included the scalability analysis of this part in the revised manuscript.

---

> > ### Author Response · Authors · 2025-11-20
> > **Rebuttal by Authors (Continued)**
> >
> > > *Why choose BNN over (sparse) GP, lightweight ensemble, or incremental update strategies? Is uncertainty estimation under ...*
> >
> > In early experiments, we initially used a Gaussian Process (GP) as the surrogate model for estimating the utility and uncertainty of feature operations. However, we found that GP (and sparse GP variants) do not scale well to the high-dimensional embeddings used to represent LLM-generated feature operations. In our setting, the input dimension is typically in the hundreds, and GP kernels were unable to capture meaningful variation across these high-dimensional representations, causing both the predicted utilities and uncertainties across candidate operations to collapse to almost identical values, which makes the acquisition function ineffective and prevented the model from distinguishing promising operations from uninformative ones. By replacing the surrogate with BNN, we observed a clear improvement. In our experiments, the BNN produced well-calibrated variations in both predicted utility and uncertainty, enabling the UCB acquisition rule to meaningfully differentiate candidate operations, which directly translated into more effective selection decisions and the performance gains reported in the evaluation section. The empirical results therefore support the choice of BNN as the surrogate model in our framework.

---

> > > ### Comment · Reviewer_M3JD · 2025-11-20
> > >
> > > The response and experimental results look good. I revised my score accordingly. good luck

---

> > > > ### Author Response · Authors · 2025-11-20
> > > >
> > > > Thank you for taking the time to reevaluate our paper. Have a great day!

---

### Official Review · Reviewer_qUc9 · 2025-11-03

**Soundness:** 3
**Presentation:** 3
**Contribution:** 3
**Rating:** 6
**Confidence:** 3

**Summary:**

This paper explores the use of Bayesian neural networks (BNN) to explicitly model the utility of feature transformations models in LLM-based feature engineering. The LLM serves as a sampler of candidate feature transformations that are then selected using a UCB bandit based on utility predictions. The authors further introduce an approach to incorporating human preference feedback in utility modeling. They demonstrate through experiments that the proposed framework improves feature engineering performance and reduces users’ cognitive load.

**Strengths:**

I think the paper suggests a valid approach to LLM-based feature engineering. The use of utility models helps selectively evaluate promising features proposed by the LLM and thus can reduce the computation cost of feature evaluations. The paper also presents an approach to incorporating human preference feedback in utility modeling. The mathematical rationale of the framework is well explained.

**Weaknesses:**

It would be appreciated if the authors could provide further details on the setting of BNNs and the optimization algorithms for solving Equations (5) and (17).

Some experimental details are missing. The standard errors across repeated runs are not provided. It is not stated how the parameters of downstream models have been selected, which could have an impact on the performance.

The experiments could evaluate other LLM backbones in addition to GPT-4o. It would be great to also include a study on the LLM generation cost and feature evaluation cost of the framework.

**Questions:**

There may be typos in line 194 and Equation (6) on $q_t(\theta))$.

I wonder why the right-hand side of inequality (11) does not involve $\mu_t(e^a_t)$ and $\mu_t(e^b_t)$.

---

> ### Author Response · Authors · 2025-11-19
> **Rebuttal by Authors**
>
> Thank you for your review! We hope the responses below satisfactorily address your concerns and questions, and we are open to discussing any further concerns you may have.
>
> > *It would be appreciated if the authors could provide further details on the setting of BNNs and the optimization algorithms for solving Equations (5) and (17).*
>
> In our implementation, the BNN is a three-layer Bayesian MLP, where all parameters are initialized by sampling from the prior distribution. the network outputs a feature operation utility estimation. To optimize this equation (5), we apply standard stochastic variational inference with the Adam optimizer. The KL term has a closed-form expression and can therefore be computed exactly. For the expected log-likelihood term, we use the reparameterization trick together with Monte Carlo sampling. In each mini-batch, we draw several samples of network parameters from the current variational distribution, perform a forward pass of the BNN for each sampled parameter set, and compute the average loss. We then backpropagate through this Monte-Carlo estimate and update all variational parameters using Adam until the loss converges.  We solve Equation (17) using a similar variational optimization procedure. However, unlike Equation (5), the model is not optimized from scratch. Instead, we take the current learned variational posterior as the initialization and perform a small number of gradient-descent updates to incorporate the additional preference-feedback likelihood.
>
> > *Some experimental details are missing. The standard errors across repeated runs are not provided. It is not stated how the parameters of downstream models have been selected, which could have an impact on the performance.*
>
> We thank the reviewer for highlighting the missing experimental details. Due to space constraints, we did not include the exact numerical standard errors for each row in Table 1. We would also like to note that the standard-error information is already visualized in Figure 1 and Figure C.1, where the shaded regions indicate the standard error of the mean across repeated runs. Regarding the downstream models, for each feature engineering method and dataset, we conduct a grid search to identify the best hyperparameters of the tabular models so that all methods are compared fairly.
>
> >*There may be typos in line 194 and Equation (6) ...*
>
> Thanks for pointing our this. We have already revised this part in the manuscript.
>
> > *I wonder why the right-hand side of inequality (11) does not involve....*
>
> We thank the reviewer for this question. Inequality (11) is obtained by
> expanding the middle term in Corollary 3.1 and then using the fact that
> Using the fact that $e_t^a$ is chosen as the UCB maximizer. Specifically:
>
> $$ \\mathrm{UCB}_t(e_t^b) - \\mathrm{LCB}_t(e_t^a) = [\\mu_t(e_t^b) + \\sqrt{\\beta_t}\\,\\sigma_t(e_t^b)] - [\\mu_t(e_t^a) - \\sqrt{\\beta_t}\\,\\sigma_t(e_t^a)] = \\mu_t(e_t^b) - \\mu_t(e_t^a) + \\sqrt{\\beta_t}(\\sigma_t(e_t^a) + \\sigma_t(e_t^b)) $$
>
>
>
> Because $e_t^a$ maximizes $\\mathrm{UCB}_t$, we have   $\\mathrm{UCB}_t(e_t^a) \\ge \\mathrm{LCB}_t(e_t^b)$, which implies that
>
> $$
> \\mu_t(e_t^b) - \\mu_t(e_t^a)
> \\le
> \\sqrt{\\beta_t}(\\sigma_t(e_t^a) + \\sigma_t(e_t^b))
> $$
>
> Subsitituting  $\\mu_t(e_t^b) - \\mu_t(e_t^a)$  with $\\sqrt{\\beta_t}(\\sigma_t(e_t^a) + \\sigma_t(e_t^b))$, we can finally get inequality 11.

---

> ### Author Response · Authors · 2025-11-20
> **Rebuttal by Authors (Continued)**
>
> >*The experiments could evaluate other LLM backbones in addition to GPT-4o...*
>
> We thank the reviewer for raising this point. To examine how the choice of backbone model affects the performance of our framework, we conducted an additional experiment using three LLMs beyond GPT-4o: DeepSeek-v3 (open-source), GPT-3.5-turbo (smaller, non-reasoning model), and GPT-5 (larger, reasoning-capable closed model). These models span different families and capability levels, allowing us to probe the stability of our approach across heterogeneous generators. Tables A and B report the average AUROC on the 13 datasets used in Table 1 of the main manuscript.
> Across all three newly added LLMs, the performance trends remain consistent with those observed in the main paper using GPT-4o. Even when the generator is relatively weak (e.g., GPT-3.5-turbo), our method effectively filters out poor or redundant feature transformations, and the overall performance stays above all baseline feature-engineering systems. Stronger generators such as GPT-5 and DeepSeek-v3 naturally yield higher-quality proposals and thus higher absolute accuracy, but the relative improvement of our method over the baselines is stable across every backbone. When a human collaborator is available, incorporating preference feedback improves performance for all generators.
> Overall, these results indicate that our feature-engineering framework is robust to the choice of backbone LLM and does not depend on any specific model family or scale.
> We have incorporated this new backbone ablation analysis into the revised manuscript.
>
> **Table B. Average AUROC on 13 datasets with different LLMs and MLP as the downstream tabular model.**
>
> | Model        | OpenFE  | AutoGluon | CAAFE  | OCTree | Ours (w/o human) | Ours (w/ human) |
> |--------------|---------|-----------|--------|--------|-------------------|------------------|
> | Deepseek-v3  | 0.8348  | 0.8345    | 0.8490 | 0.8550 | 0.8610            | 0.8640           |
> | GPT3.5-turbo | 0.8348  | 0.8345    | 0.8315 | 0.8420 | 0.8460            | 0.8510           |
> | GPT4o        | 0.8348  | 0.8345    | 0.8430 | 0.8470 | 0.8530            | 0.8550           |
> | GPT5         | 0.8348  | 0.8345    | 0.8554 | 0.8575 | 0.8590            | 0.8651           |
>
>
> **Table B. Average AUROC on 13 datasets with different LLMs and XGBoost as the downstream tabular model.**
>
> | Model        | OpenFE  | AutoGluon | CAAFE  | OCTree | Ours (w/o human) | Ours (w/ human) |
> |--------------|---------|-----------|--------|--------|-------------------|------------------|
> | Deepseek-v3  | 0.8558  | 0.8535    | 0.8660 | 0.8731 | 0.8820            | 0.8860           |
> | GPT3.5-turbo | 0.8558  | 0.8535    | 0.8521 | 0.8603 | 0.8654            | 0.8710           |
> | GPT4o        | 0.8558  | 0.8535    | 0.8590 | 0.8670 | 0.8740            | 0.8740           |
> | GPT5         | 0.8558  | 0.8535    | 0.8712 | 0.8774 | 0.8801            | 0.8870           |

---

> ### Author Response · Authors · 2025-11-20
> **Rebuttal by Authors (Continued)**
>
> > *It would be great to also include a study on the LLM generation cost and feature evaluation cost of the framework.*
>
> To analyze the computational cost for each component of  the proposed method, we measured the runtime of each component in a single feature-engineering iteration. Each round of our method consists of four steps:
>
> 1. calling the LLM to generate candidate feature transformations,
> 2. fitting the BNN surrogate model using the accumulated observations,
> 3. computing the UCB score for each proposed candidate, and
> 4. evaluating the selected transformation using the downstream tabular model.
>
>
>  Among these components, steps 2–3 correspond to the core contribution of our method, while steps 1 and 4 are shared by all LLM-based feature engineering pipelines.
> To avoid confounding effects from dataset semantics, we constructed synthetic binary classification datasets and systematically varied (a) the number of initial feature columns and (b) the number of data instances to directly measure how each component of the pipeline scales with feature dimensionality and dataset size. All feature columns were generated from standard Gaussian distributions, labels from a Bernoulli distribution, an MLP model was used as the downstream evaluator, and all LLM calls were made using the GPT-4o API.
> We first fixed the dataset size to 10,000 samples and varied the number of initial feature columns from 10 to 10,000. As shown in Table A, the surrogate model fitting time and UCB score computation time increase only mildly as the number of features grows, while the downstream model evaluation grows more noticeably. The LLM call time remains constant, since it depends only on the prompt size rather than the data size. Even with as many as 10,000 initial features, the surrogate and UCB components together remain about 2.2% of the total runtime.
>
> **Table A. Runtime breakdown when varying the number of initial features (n_samples = 10,000).**
>
> | #Features | LLM Mean (s) | Surrogate Mean (s) | UCB Mean (s) | Eval Mean (s) |
> |-----------|--------------|---------------------|--------------|----------------|
> | 10        | 1.820303     | 0.166093            | 0.006227     | 1.796845       |
> | 50        | 1.820303     | 0.162171            | 0.004734     | 1.247297       |
> | 100       | 1.820303     | 0.190383            | 0.004638     | 1.781571       |
> | 1,000     | 1.820303     | 0.201621            | 0.008805     | 8.040899       |
> | 10,000    | 1.820303     | 0.570452            | 0.018869     | 23.431735      |
>
> Next, we fixed the number of initial features to 100 and varied the dataset size from 1,000 to 100,000 instances. As shown in Table B, the surrogate fitting and UCB computation times remain nearly constant across all sample sizes and contribute only a small percentage of the total runtime, because both operate at the feature-operation level and are independent of the number of rows. In contrast, the downstream evaluation time increases with dataset size, as it requires training the MLP on the full dataset. For example, even at 100,000 samples, the surrogate and UCB components together account for only about 1.3 percent of the total computation.
>
> The LLM call time again remains constant.
>
> **Table B. Runtime breakdown when varying the number of samples (n_features = 100).**
>
> | #Samples | LLM Mean (s) | Surrogate Mean (s) | UCB Mean (s) | Eval Mean (s) |
> |----------|--------------|---------------------|--------------|----------------|
> | 1,000    | 1.820303     | 0.173182            | 0.005338     | 0.289455       |
> | 5,000    | 1.820303     | 0.179113            | 0.005404     | 0.887437       |
> | 10,000   |1.820303     | 0.232185            | 0.006151     | 1.465134       |
> | 50,000   | 1.820303     | 0.182320            | 0.005778     | 5.222195       |
> | 100,000  | 1.820303     | 0.179800            | 0.005241     | 10.651274      |
>
> Taken together, these results show that our method scales effectively in both directions.
> When fixing the number of features and increasing sample size from 1,000 to 100,000, the surrogate model  plus UCB components consistently contribute only about 1–2 percent of total runtime.
> When fixing sample size and increasing the number of features from 10 to 10,000, they remain below 3 percent, while the downstream model evaluation accounts for more than 95 percent of total cost.
> Overall, the core components introduced by our method add minimal extra computation cost and scale smoothly to large datasets and high-dimensional feature spaces.  We also included the scalability analysis of this part in the revised manuscript.

---

### Author Response · Authors · 2025-11-30
**Summary of Revisions and Responses**

We would like to begin by sincerely thanking the reviewers for their careful review and for their constructive engagement throughout both the initial evaluation and the rebuttal period. We also greatly appreciate the time and effort that the new Area Chair is dedicating to reevaluating our submission under this year’s exceptional circumstances.

During the prior rebuttal phase, we carefully clarified some misunderstandings and addressed the issues raised across the reviews.  We were encouraged to see that, during the discussion, reviewers indicated that our clarifications and the additional results had addressed the major concerns they initially raised or had slightly misunderstood in their comments, and that this led them to hold a more positive view of the submission and update their ratings accordingly. In particular, both Reviewer abCY and Reviewer M3JD, who had initially expressed more critical assessments, indicated during the discussion phase that the additional analyses and clarifications resolved the issues they had identified. Below, we summarize the key points from our rebuttal for the Area Chair’s reference, in the hope of facilitating your evaluation.

> 1. Novelty and conceptual contribution. (Reviewer abCY)

One reviewer initially questioned the novelty of our method and misunderstood our porposed method,  assuming the LLM in our method directly evaluated feature transformations. In the rebuttal, we clarified that the LLM only proposes candidates, while selection is guided by a Bayesian surrogate that models both utility and uncertainty of LLM-proposed feature operations to fully decouple proposal and selection, unlike prior methods such as CAAFE and OCTree. This design yields substantial improvement over prior non-LLM-based methods and LLM-based methods, even without human feedback. We further explained how selective preference feedback alleviates early-stage surrogate underfitting.

> 2. Sensitivity to LLM backbones, anonymized columns, and missing details. (Reviewers qUc9, M3JD, oY9U, abCY)

 Reviewers asked whether weaker LLM generators would degrade the framework's performance and whether the method works without semantic column names. We added a backbone ablation with DeepSeek-v3, GPT-3.5-turbo, GPT-4o, and GPT-5, showing that across all backbones, both variants of our method consistently outperform baselines, with stable relative gains. Candidate-quality analysis confirmed zero label leakage and showed that weaker models generate simpler, more repetitive transformations, while stronger models produce more compositional ones. We also conducted anonymized-column experiments and showed our method remains effective and superior to CAAFE and OCTree. Finally, we explained missing details, such as the configuration of the BNN and downstream hyperparameters.

> 3. Scalability and extra computational cost. (Reviewers qUc9, M3JD, oY9U, abCY)

 Reviewers asked how our method scales to large-column or large-sample datasets, and whether the proposed method are computationally expensive. We added a comprehensive runtime study varying the number of initial features (10 → 10,000) and samples (1,000 → 100,000). Across all settings, surrogate model fitting + UCB remain only 1–3% of total runtime, while downstream model evaluation dominates. LLM call time stays constant because it depends solely on prompt size. We emphasized that human experts never inspect the full raw feature space as they only compare two proposed transformations, so human scalability is not affected by the number of columns too much.

> 4. Human feedback, simulated experts, and robustness to expertise levels. (Reviewers M3JD, oY9U)

 Reviewers raised concerns about using GPT-4o as a simulated expert, the noise characteristics of such feedback, the choice of $\gamma$, and the impact of fatigue and latency on the choice of $\gamma$ . We introduced new robustness experiments injecting synthetic noise (flip probabilities 0.1–0.5), showing that our method remains stable and consistently outperforms baselines even under significant noise. We also connected these results to our real-user study, where ML practitioners achieved strong AUROC.
We clarified that $\gamma$ was set via a pilot study and is tunable across expert groups. Importantly, our method queries human preferences only selectively (~7 times per 50 iterations) for  scalability of human effort.

In our revised manuscript, we have incorporated the new ablation study across different backbone LLM models, the full scalability analysis of our method,  corrected minor typos, and added the missing citations such as P2T and STUNT. We are grateful for the reviewers’ thoughtful feedback, which helped us improve the clarity and rigor of the work. We also sincerely appreciate the Area Chair’s effort in evaluating our responses and the revised manuscript, and we hope this summary helps facilitate your assessment.

---

### Meta-Review · Area_Chair_y2f9 · 2026-01-08

**Summary:**

Among the four reviewers, two provided positive scores and two provided negative scores. All reviewers acknowledge that the proposed framework is effective and that the paper is clearly written. The work presents a human–LLM collaboration framework: specifically, LLMs are used only to generate candidates, while a surrogate model guided by Bayesian optimization is used to model the utility and uncertainty of these candidates. Since accurately estimating utility is often difficult, the framework selectively solicits input from human experts.

The two negative reviewers’ main concerns are (i) whether it is reasonable and appropriate to use a GPT-4o-based model as a substitute for human preferences, and (ii) whether the work about human preferences offers sufficient novelty. In contrast, the two positive reviewers fully endorse the framework’s effectiveness and potential impact.

**Reviewer Concerns:**

In my view, the rebuttal addresses the reviewers’ concerns well. The authors’ key clarification is that incorporating human preferences is only one component of the overall framework. The primary contribution is the Bayesian surrogate modeling of each candidate’s utility and uncertainty (within the Bayesian-optimization loop), while the use of human preference feedback is intended as a complementary element.

That said, the specific setup and implementation details of incorporating human preferences remain debatable and could still be questioned. Nevertheless, I believe the framework itself is sufficiently novel and promising in terms of both its methodological contribution and potential impact.

**Reviewer Scores:**

In my view, the authors’ rebuttal is sincere and addresses the feedback in a concrete and persuasive manner, both methodologically and empirically. I expect that the two negative reviewers would correspondingly increase their scores, as they themselves indicated in their follow-up responses.

---

### Decision · Program_Chairs · 2026-01-26

Accept (Poster)